# The yeast mitophagy receptor Atg32 is ubiquitinated and degraded by the proteasome

Nadine Camougrand[1]*, Pierre Vigié[1], Cécile Gonzalez[1¤], Stéphen Manon[1], Ingrid Bhatia-Kiššová[2]

1 CNRS and Université de Bordeaux, IBGC, UMR5095, Bordeaux, France, 2 Department of Biochemistry, Faculty of Natural Sciences, Comenius University, Bratislava, Slovak Republic

¤ Current address: IBCP, UMR5086, Université de Lyon, Lyon, France
* n.camougrand@ibgc.cnrs.fr

## Abstract

Mitophagy, the process that degrades mitochondria selectively through autophagy, is involved in the quality control of mitochondria in cells grown under respiratory conditions. In yeast, the presence of the Atg32 protein on the outer mitochondrial membrane allows for the recognition and targeting of superfluous or damaged mitochondria for degradation. Post-translational modifications such as phosphorylation are crucial for the execution of mito-phagy. In our study we monitor the stability of Atg32 protein in the yeast *S. cerevisiae* and show that Atg32 is degraded under normal growth conditions, upon starvation or rapamycin treatment. The Atg32 turnover can be prevented by inhibition of the proteasome activity, suggesting that Atg32 is also ubiquitinated. Mass spectrometry analysis of purified Atg32 protein revealed that at least lysine residue in position 282 is ubiquitinated. Interestingly, the replacement of lysine 282 with alanine impaired Atg32 degradation only partially in the course of cell growth, suggesting that additional lysine residues on Atg32 might also be ubi-quitinated. Our results provide the foundation to further elucidate the physiological signifi-cance of Atg32 turnover and the interplay between mitophagy and the proteasome.

**Data Availability Statement:** All relevant data are available on BioRxiv 652933: https://doi.org/10.1101/652933.

## Introduction

Mitochondria are organelles in charge of many crucial functions in cells. In eukaryotes, they are known to be the powerhouse of cells by producing ATP via oxidative phosphorylation; they are also involved in different synthesis pathways such as the synthesis of certain amino acids and lipids and in signaling pathways, such as apoptosis and calcium. Mitochondrial dys-functions have been associated with aging and with an increasing number of pathologies, including neurodegenerative diseases, cancer, and metabolic perturbation. To maintain cell survival and homeostasis, cells have developed ways to remove superfluous or damaged cell components and organelles such as mitochondria. Mitophagy is one of the processes involved in mitochondrial quality control. Mitophagy is a selective form of autophagy [1, 2]. Macroau-tophagy (hereafter called autophagy) is conserved among eukaryotic species and involves

**Funding:** IBK: Slovak APVV agency (SK-FR-2015-0005) NC and PV: CNRS, the University of Bordeaux and the Doctoral International Program from IDEX of Bordeaux supported by Agence Nationale pour la Recherche (ANR-10-IDEX-03-02) NC and IBK: France/Slovakia collaboration was supported by Campus France (Stefanik n˚ 35809VC) The funders had no role in study design, data collection and analysis, decision to publish, or preparation of the manuscript.

**Competing interests:** The authors have declared that no competing interests exist.

specific lytic compartments: the vacuole in yeast and lysosomes in mammals. It also involves specific autophagy-related (Atg) proteins. To date, more than 40 Atg proteins have been identified, about half are involved in the core autophagy machinery, and the rest are implicated in specific autophagy processes and regulations.

In mammalian cells, mitophagy and recognition of the mitochondria that need to be removed are complex processes. Two main mechanisms of mitophagy signaling have been identified: ubiquitin-mediated mitophagy and receptor-mediated mitophagy. To be degraded by mitophagy, mitochondria must be labeled with a flag that is recognized by autophagic machinery. For these mitophagy-signaling pathways, ubiquitin/proteins, receptors/adaptors containing the LIR (LC3-interacting region) domain act to recruit LC3 family members for mitochondria and thus induce mitophagy.

In yeast, Atg32 is the only characterized mitophagy receptor, which suggests mitophagy could be a simpler mechanism in yeast than in mammals [3, 4]. The Atg32 protein is an outer mitochondrial membrane protein with its C-terminus in the intermembrane space and its N-terminus in the cytosol [3, 4]. As mammalian protein receptors, Atg32 possesses the Atg8-interacting motif (AIM, LIR domain). When mitophagy is induced, Atg32 interacts with Atg11, a cytosolic adaptor protein required for selective autophagy. Atg11 is believed to target mitochondria to the pre-autophagosomal structure (PAS), where an autophagosome is generated to enclose the mitochondria. At the PAS, Atg32 interacts with Atg8, a protein anchored in autophagosome membranes, and the Atg32-Atg8 interaction facilitates the formation of autophagosomes surrounding mitochondria [3–6]. Both interactions are required for the recruitment of mitochondria to the phagophore followed by their sequestration within autophagosomes and their degradation in the vacuole.

The Atg32 expression is regulated by multiple factors. It has been shown that N-acetylcysteine (NAC), which increases the cellular level of reduced glutathione (GSH), inhibits mitophagy [7] through suppression of Atg32 expression [4]. Indeed, the absence of an Opi3 protein, an enzyme required in the conversion of phosphatidylethanolamine to phosphatidylcholine, causes an increase in intracellular reduced glutathione, leading to a decrease in Atg32 protein expression [8]. Moreover, the absence of some components of the NatA N-acetyltransferase complex also results in a decrease in Atg32 protein expression [9]. However, the precise role of NatA concerning Atg32 protein remains unclear. In addition, it has been shown that Atg32 phosphorylation on serines 114 and 119, is required for mitophagy activation [10, 11]. More recently, Levchenko et al. detected another post-translational modification of Atg32 when mitophagy was induced by rapamycin treatment in a *pep4Δ* background, where vacuolar proteolysis is impaired. However, its precise nature remains to be characterized [12].

Because mitophagy induction is highly regulated, we reasoned, as a key mitophagy factor, Atg32 protein may be subject to degradation control. In this study, we monitored the expression and the stability of the Atg32 protein during mitophagy in the yeast *S. cerevisiae*. We showed the existence of an interplay between mitophagy and the proteasome. We demonstrated that a novel post-translational modification of Atg32, ubiquitination at least on lysine 282 residue, modulated Atg32 protein level during the cell growth. Because uncontrolled mitophagy activation can lead to dire consequences, understanding Atg32 regulation would help us understand the control of mitophagy activity and also the cooperation between the proteasome and mitophagy.

## Material and methods

### Yeast strains, plasmids and growth conditions

All yeast strains used in this study are listed in the S1 Table and derived from BY4742 (Euroscarf bank). Yeast cells were grown aerobically at 28˚C in complete minimal synthetic medium

(CMS; 0.175% yeast nitrogen base without amino acids and ammonium sulfate, 0.5% ammonium sulfate, 0.1% potassium phosphate, 0.2% Drop-Mix, 0.01% of auxotrophic amino acids and nucleotide, pH 5.5), supplemented with 2% lactate as a carbon source (CMS-L). Cell growth was followed by optical density at 600 nm. Atg32 expression was examined in cells harvested in the mid-exponential phase of growth (1.5–2 $OD_{600}$; T0), after 8 h (8 h; late exponential phase), after 1 day (24 h; early stationary phase), and after 2 days (48 h; late stationary phase) of culture. To inhibit the proteasome, 75 μM MG-132 + 0.003% SDS were added to the cell culture at 8 h time point. To inhibit vacuolar proteolysis, 2 mM PMSF was added to the cell culture at 8 h time point; this step was repeated twice during the course of cell growth. For starvation experiments, cells were harvested at the mid-exponential phase of growth (1.5–2 $OD_{600}$; T0), washed three times with water and incubated in a nitrogen starvation medium (-N; 0.175% yeast nitrogen base without amino acids and ammonium sulfate, and 2% lactate, pH 5.5) for 3-, 6- or 24 h (-N3h, -N6h, -N24h). The *pre2-2* mutant strain was unable to grow in the lactate-containing medium; for this reason, the experiments including *pre2-2* mutant strain were performed in a CMS medium supplemented with 2% galactose as a carbon source (CMS-G). The *pre2-2* mutant strain was a gift of Dr. Sagot (UMR5095 CNRS, France).

Mitophagy was studied in cells expressing the mitochondrial matrix Idp1-GFP protein, and subjected to nitrogen starvation or during different phases of growth. The plasmid p*IDP1-GFP* was a gift from Dr. Abeliovich (Hebrew University of Jerusalem, Israel). To study autophagy, cells were transformed with plasmid *pGFP-ATG8* kindly provided by Dr. Dan Klionsky (Michigan University, USA). The plasmid expressing HA-Atg32 from copper promoter was a gift from Dr. Dan Klionsky. The plasmid expressing Atg32-V5-6HIS from *ATG32* promoter was a gift from Axel Athané (UMR5095 CNRS, France). To prepare the pPROM-*ATG32-β-galactosidase* plasmid, the promoter of *ATG32* gene was amplified by PCR using forward primer (5'-GTGATGTATCCACAGGGAATTCCGCTC-3') and reverse primer (5'-CTTTTAGATGAGGATCCTTTACCT-3'), and cloned into *Bam*H1-digested YEp357 plasmid kindly provided by Dr. Pinson (UMR5095 CNRS, France). The plasmid *pYES2-ATG32-V5* expressing mutated Atg32[K282A]-V5 or Atg32-AAAA-V5 proteins were generated using the QuikChange Site-Directed Mutagenesis approach with the following primers: K282A forward (5'-CAATATTCTCAAGGCGCGCCTGTAATA-3'), K282A reverse (5'-GATCGGTATTACAGGCGCGCCTTGAGA-3)'; Rsp5-binding motif forward (5'- AAAG AATACCAATCTCTTTTTGAAGCAQGCGGCAGCTCACGATTCCGCAACATTC-3'); Rsp5-binding motif reverse (5'-TTGCGGGAATGTTGCGGAATCGTGAGCTGCCGCTGCTTCAA AAAGAGATTGGTA-3').

## Preparation of protein extracts and western blots

For preparation of total protein extracts, 2 x $10^7$ cells were harvested by centrifugation, washed with water, and resuspended in 450 μl of water and 50 μl of lysis buffer (1.85 M NaOH, 3.5% β-mercaptoethanol). After 10 min on ice, 50 μl of trichloroacetic acid 3 M was added followed by another incubation of 10 min on ice. Proteins were pelleted by centrifugation, 8 min at 13 000 *g*, washed with acetone and resuspended in 20 μl of 5% SDS and 20 μl of loading buffer (2% *β*-mercaptoethanol, 2% SDS, 0.1 M Tris-HCl, pH 8.8, 20% glycerol, 0.02% bromophenol blue). Samples were boiled for 5 min and 50 μg of proteins/total protein extracts corresponding to 5 x $10^6$ cells per line were separated by electrophoresis by 12.5% SDS-PAGE and subjected to immunoanalysis with either anti-GFP (Roche), anti-Pgk1 (Invitrogen), anti-porin (Invitrogen), anti-Dpms1 (Invitrogen), anti-HA (Roche), anti-V5 (Invitrogen), anti-HIS (Euromedex) and anti-ubiquitin (Calbiochem) antibodies. Detection was performed with ECL[+] reagent (Luminata Forte, Perkin Elmer). Pgk1, the cytosolic phosphoglycerate kinase,

was used as a loading control. The Atg32-V5/Pgk1, HA-Atg32/Pgk1 or GFP/GFP + Idp-GFP ratios were quantified by using ImageJ software (NIH). Results of calculations were expressed as the mean ± SEM. To study Atg32-V5 turnover, between 5 and 8 independent experiments were carried out for each tested condition; one representative western blot result is presented for each experimental condition. *P*-values were assessed using paired Student's t-tests; $^*$ $P < 0.05$ or $^{**}$ $P < 0.01$ were considered statistically significant.

For preparation of cell lysates, $5 \times 10^7$ cells were broken with glass beads in a buffer containing 0.6 M sorbitol, 20 mM MES pH 6 plus protease inhibitor cocktail (Roche); lysates were centrifuged 10 min at 800 *g*. The supernatants, an equivalent of 400 μg of proteins, were loaded on 20–55% OptiPrep density gradients in a buffer with 0.6 M sorbitol 20 mM MES pH6, 5 mM EDTA, pH6 plus protease inhibitor cocktail (Roche). Fractions were collected, precipitated with TCA and pellets were resuspended in 20 μl of 5% SDS and 20 μl of loading buffer (2% β-mercaptoethanol, 2% SDS, 0.1 M Tris-HCl, pH 8.8, 20% glycerol, 0.02% bromophenol blue). Samples were boiled for 5 min and 4 μl of samples were separated by 12.5% SDS-PAGE and subjected to immunoanalysis.

## Monitoring mitophagy by western blot

BY4742, *atg32Δ* mutant cells and *atg32Δ* cells expressing Atg32-V5 and the mitochondrial matrix Idp1-GFP were grown in a CMS-L medium in presence or absence of MG-132 or exposed to nitrogen starvation, respectively. At different times (as indicated in each figure), total protein extracts were prepared from $2 \times 10^7$ cells as described above. Extracts were immunoblotted, 12.5% SDS-PAGE gels were used. Under basal condition (T0), only the Idp1-GFP form (75 kDa) was detected with a monoclonal mouse anti-GFP antibody. Upon mitophagy induction, Idp1-GFP (75 kDa) and free GFP (27 kDa) were detected with anti-GFP antibodies. Pgk1 was used as a loading control. Quantification of GFP signals was done using ImageJ software by calculating the ratio of free GFP to total GFP (GFP/GFP + Idp1-GFP). Between 4 and 5 independent experiments were carried out for each experimental condition; one representative western blot result is presented.

## Monitoring autophagy by western blot

BY4742 and *atg32Δ* cells expressing Atg32-V5 and GFP-Atg8 were grown in a CMS-L medium in presence or absence of MG-132. At different times (as indicated in each figure), total protein extracts were prepared from $2 \times 10^7$ cells as described above. Extracts were immunoblotted. Under basal condition (T0), only the GFP-Atg8 form (40.6 kDa) was detected with anti-GFP antibody. Upon autophagy induction, GFP-Atg8 localized in autophagosome membranes is delivered into vacuoles, here the proteolysis of GFP-Atg8 releases an intact GFP moiety (27 kDa) that can be detected with anti-GFP antibodies. Pgk1, was used as a loading control. Three independent experiments were carried out for each examined condition; one representative result is shown.

## ALP activity

Alkaline phosphatase activity was measured on *pho8Δ* strain, expressing the mitochondria-targeted truncated version of Pho8, mtPho8Δ60 [13], during the mid-exponential (T0) and the stationary phase (48 h) in absence or presence of MG-132. For each point, $5 \times 10^7$ cells expressing mitochondria-targeted Pho8Δ60 (mtPho8Δ60) were harvested and lysed with glass beads in lysis buffer (20 mM PIPES, 0.5% Triton X-100 0.5%, 50 mM KCl, 100 mM potassium acetate, 10 mM $MgCl_2$, 10 μM $ZnSO_4$, 2 mM PMSF). After centrifugation, 20 μl of supernatant were put with 80 μl of water and 400 μl of activity buffer (250 mM Tris HCl pH 8, 0.4% Triton

X-100, 10 mM $MgCl_2$, 10 μM $ZnSO_4$, 125 mM p-nitrophenyl-phosphate) during 20 minutes at 30°C. The reaction was stopped by the addition of 500 μl of 1 M glycine pH 11. Activity was then measured by optical density at 400 nm. Protein concentration was measured with the Lowry method. ALP activities were expressed as arbitrary fluorescence units/minute/mg proteins (AU/min/mg). 4 independent experiments were carried out for each condition. Results of quantification were expressed as the mean ± SEM. $P$-values were assessed using paired Student's t-tests; ** $P < 0.01$ were considered statistically significant.

### β−galactosidase activity

BY4742 cells, expressing $\beta$-galactosidase protein under control of *ATG32* promoter were grown in a lactate-containing medium and harvested at different time points: mid-exponential phase of growth (T0), late exponential phase of growth (8 h), early stationary phase of growth (24 h) and late stationary phase of growth (48 h). For proteasome inhibition, 75 μM MG-132 + 0.003% SDS were added at 8 h time point. To measure $\beta$-galactosidase activity, five $OD_{600}$ (corresponding to $5.0 \times 10^7$ cells) were harvested and broken with glass beads in lysis buffer (100 mM Tris-HCL pH 8, 1 mM DTT, 20% glycerol, 2 mM PMSF) during 4 min at 20 hertz using Mixer Mill MM400 (Retsch). Then, 20 μl of cell lysate were used to measure $\beta$-galactosidase activity by absorbance at 420 nm in 60 mM $Na_2HPO_4$, 40 mM $NaH_2PO_4$, 10 mM KCl, 1 mM $MgSO_4$, 50 mM β-mercaptoethanol, pH 7 and with 0.8 mg/ml of ortho-Nitrophenyl-β-galactoside (ONPG). Protein concentrations were measured using Lowry's method and activities were normalized to T0. Between 5 and 8 independent experiments were carried out for each experimental condition. Results of quantification were expressed as the mean ± SEM. $P$-values were assessed using paired Student's t-tests; ** $P < 0.01$ were considered statistically significant.

### Atg32 purification

To purify Atg32 protein, $20 \times 10^7$ cells expressing Atg32-V5-6xHIS and grown until late stationary phase (48 h) in the presence of 75 μM MG-132 were collected, and lysates were prepared as described in Becuwe et al. [14]. Cells were precipitated with TCA to a final concentration of 10% overnight at 4°C, and lysed with glass beads for 20 min at 4°C. After centrifugation, the lysate was collected and centrifuged at 16,000 *g* for 10 min. The pellet was resuspended with 30 μl of 1 M non buffered Tris and 200 μl of guanidinium buffer (6 M GuHCl, 20 mM Tris-HCl, pH 8, 100 mM $K_2HPO_4$, 10 mM imidazole, 100 mM NaCl, and 0.1% Triton X-100) and incubated for 1 h at room temperature on a rotating platform. After centrifugation (16,000 *g* for 10 min at room temperature), the lysate was loaded on a HisTrap FF Crude column (GE Healthcare Life Sciences). The column was then washed with a speed of 1 ml/min with guanidinium buffer, then with wash buffer 1 (20 mM Tris-HCl, pH 8.0, 100 mM $K_2HPO_4$, 20 mM imidazole, 100 mM NaCl, and 0.1% Triton X-100) and then with wash buffer 2 (20 mM Tris-HCl, pH 8.0, 100 mM $K_2HPO_4$, 10 mM imidazole, 1 M NaCl, and 0.1% Triton X-100). His6-tagged proteins were finally eluted with an elution buffer (50 mM Tris-HCl, pH 8.0, and 250 mM imidazole). Fractions absorbing at 254 nm were pooled, analyzed by 11% SDS-PAGE and submitted to blue colloidal staining and immuno-analysis.

### Sample preparation and protein digestion for MS analysis

After colloidal blue staining, both obtained bands (B1, B2) were cut out from the SDS-PAGE gel and subsequently cut in 1 mm x 1 mm gel pieces. Gel pieces were destained in 25 mM ammonium bicarbonate, 50% acetonitrile (ACN), rinsed twice in ultrapure water and shrunk in ACN for 10 min. After ACN removal, gel pieces were dried at room temperature, covered

with the trypsin solution (10 ng/μl in 50 mM $NH_4HCO_3$), rehydrated at 4˚C for 10 min, and finally incubated overnight at 37˚C. Spots were then incubated for 15 min in 50 mM $NH_4HCO_3$ at room temperature with rotary shaking. The supernatant was collected, and an $H_2O$/ACN/HCOOH (47.5:47.5:5) extraction solution was added onto gel slices for 15 min. The extraction step was repeated twice. Supernatants were pooled and concentrated in a vacuum centrifuge to a final volume of 100 μl. Digests were finally acidified by the addition of 2.4 μl formic acid (5%, v/v) and stored at -20˚C.

## nLC-MS/MS analysis

Peptide mixture was analyzed on the Ultimate 3000 nanoLC system (Dionex, Amsterdam, The Netherlands) coupled to the Electrospray Orbitrap Fusion™ Lumos™ Tribrid™ Mass Spectrometer (Thermo Fisher Scientific, San Jose, CA). Ten microliters of peptide digests were loaded onto a 300-μm-inner diameter x 5-mm $C_{18}$ PepMap™ trap column (LC Packings) at a flow rate of 10 μL/min. The peptides were eluted from the trap column onto an analytical 75-mm id x 50-cm C18 Pep-Map column (LC Packings) with a 4–40% linear gradient of solvent B in 45 min (solvent A was 0.1% formic acid and solvent B was 0.1% formic acid in 80% ACN). The separation flow rate was set at 300 nL/min. The mass spectrometer operated in positive ion mode at a 1.8-kV needle voltage. Data were acquired using Xcalibur 4.1 software in a data dependent mode. MS scans (*m/z* 375–1500) were recorded at a resolution of R = 120 000 (at m/z 200) and an AGC target of 4 x $10^5$ ions collected within 50 ms. Dynamic exclusion was set to 60 s and top speed fragmentation in HCD mode was performed over a 3 s cycle. MS/MS scans with a target value of 3 x $10^3$ ions were collected in the ion trap with a maximum fill time of 300 ms. Additionally, only +2 to +7 charged ions were selected for fragmentation. Other settings were as follows: no sheath nor auxiliary gas flow, heated capillary temperature, 275˚C; normalized HCD collision energy of 30% and an isolation width of 1.6 m/z. Monoisotopic precursor selection (MIPS) was set to Peptide and an intensity threshold was set to 5 x $10^3$.

## Results

### Amount of the mitophagy receptor Atg32 decreases during the stationary phase

Mitophagy in yeast is induced during nitrogen starvation, during the stationary phase of growth, or through rapamycin treatment, where cells are grown in a strictly respiratory carbon source medium. To study Atg32 expression in cells grown in a lactate-containing medium (CMS-L), we C-terminally tagged the Atg32 protein with a V5-6HIS epitope (hereafter referred to as Atg32-V5), and the tagged protein was expressed under the control of its promoter. First, we ensured the fusion protein Atg32-V5 was localized to mitochondria by analyzing total cellular extracts from cells harvested in the mid-exponential growth phase on a 20–55% OptiPrep gradient (Fig 1A) or isolated mitochondria on a 20–60% sucrose gradient (S1A Fig). In both cases, Atg32-V5 is colocalized with porin, an outer mitochondrial membrane protein. Moreover, by using mitophagy-dependent processing of the Idp1-GFP technique [15] we showed that Atg32-V5 protein was able to reverse the mitophagy defect of the *atg32Δ* mutant strain in both the stationary phase of cell growth (Fig 1B) as well as during starvation (S1B Fig). These observations align with the data of Levchenko *et al.* (2016), who showed that the C-terminal tagging of Atg32 by a ZZ tag did not interfere with mitophagy induced by nitrogen starvation or rapamycin treatment [12].

We then determined Atg32 expression from cells harvested in the mid-exponential phase of growth (1.5–2 $OD_{600}$; T0), after 8 hours (late exponential phase; 8 h), after 1 day (early

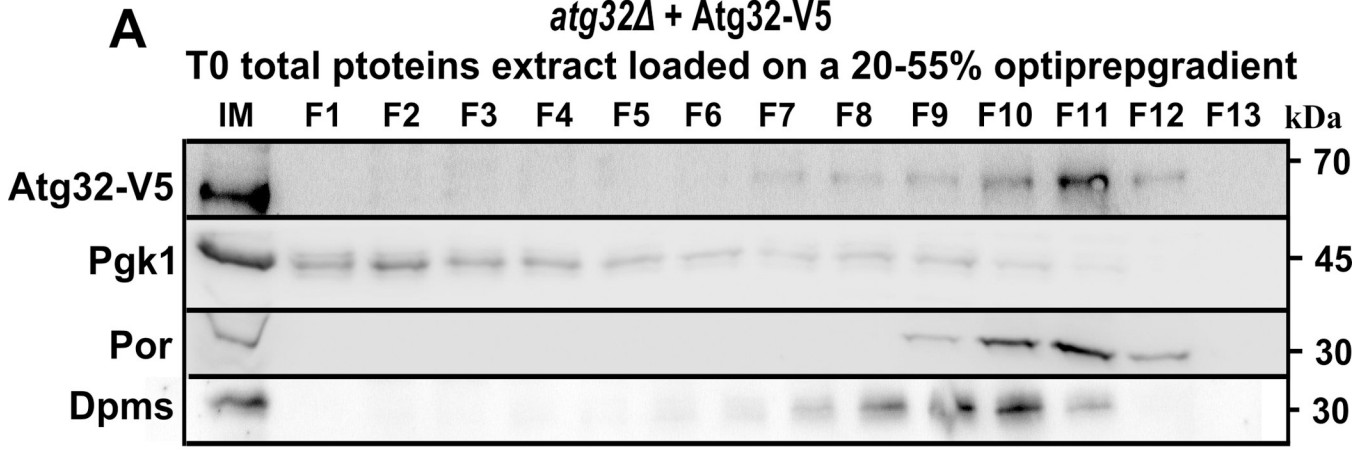

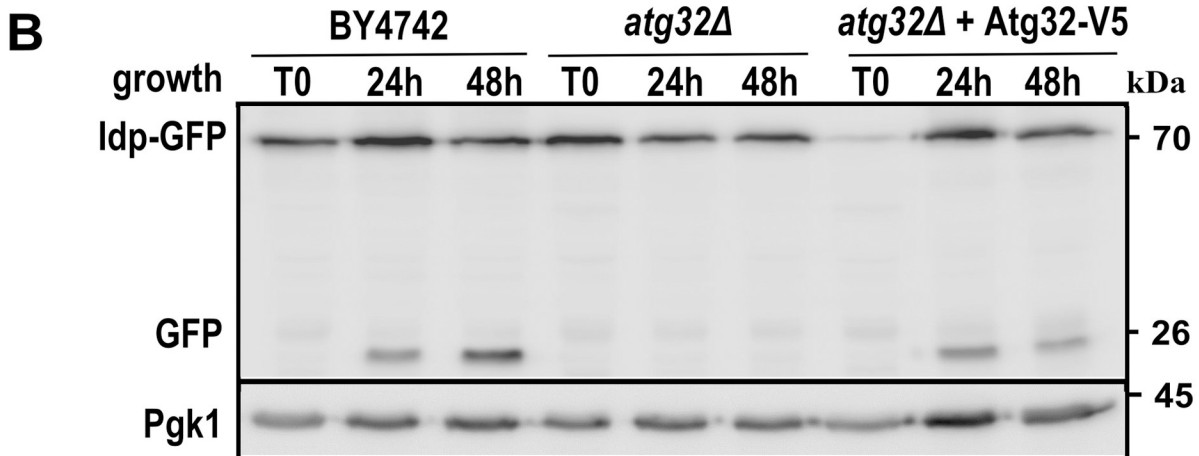

**Fig 1. Atg32-V5 protein localizes to mitochondria and restores mitophagy in *atg32Δ* mutant strain.** (A) *atg32Δ* mutant cells grown in a CMS-L medium and expressing Atg32-V5 fusion protein were harvested in a mid-exponential phase of growth (T0). Then cells were lysed and lysates were separated in a 20–55% OptiPrep density gradient. Fractions were collected after centrifugation and analyzed by western blots. Anti-V5 antibody was used to visualize Atg32-V5, and anti-Porin antibody was used to detect mitochondria-containing fractions. Pgk1, the cytosolic phosphoglycerate kinase, was used as a cytosolic marker; and Dpms, dolichyl phosphate mannose synthase, was used as an endoplasmic reticulum marker. **(B)** Mitophagy was assessed using mitophagy-dependent processing of the Idp1-GFP tool in wild-type (BY4742), *atg32Δ* mutant, or *atg32Δ*-expressing Atg32-V5. Cells were harvested in a mid-exponential phase of growth (T0), an early stationary phase (24 h), or a late stationary phase (48 h). The total protein extracts corresponding to 5 x 10$^6$ cells/50 ug proteins per line were separated by 12.5% SDS-PAGE and analyzed by immunodetection using anti-GFP antibody.

stationary phase; 24 h), and after 2 days (late stationary phase; 48 h) of culture. We found that under respiratory conditions, the Atg32-V5 protein level decreased during cell growth and almost completely disappeared in the stationary phase (Fig 2A and 2B). Similarly, we observed that when the Atg32 protein was N-terminally tagged with an HA epitope, its amount also decreased gradually during cell growth into the stationary phase (Fig 2C and 2D). It is evident that mitophagy alone cannot be responsible for the decreasing amount of Atg32 protein, as cells in the stationary phase still contain mitochondria, and mitochondrial functions are essential for their viability.

At the same time, during the first 3 hours of nitrogen starvation, Atg32-V5 levels dropped by 40% on average, changing only minimally and steadying at about 50% of the original

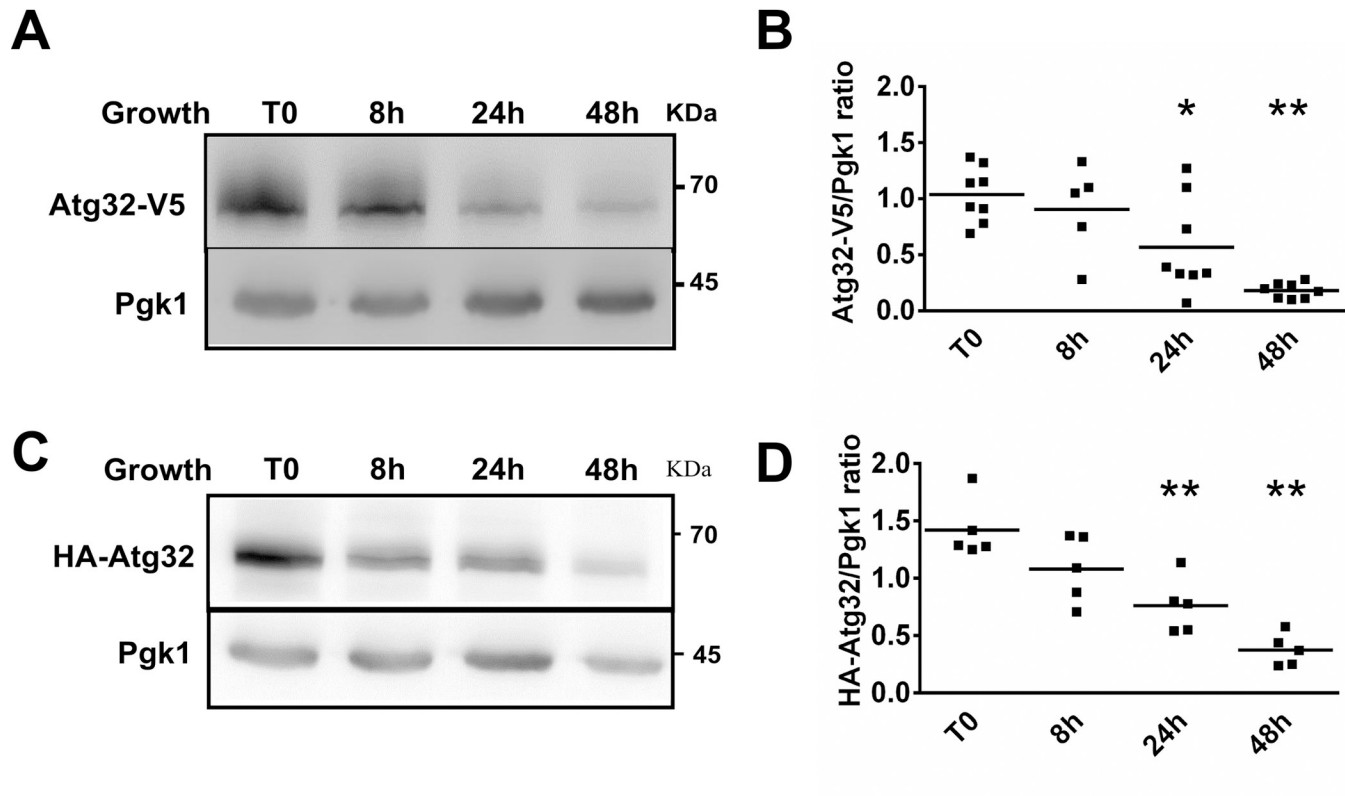

**Fig 2. The Atg32 protein is degraded during cell growth.** a*tg32Δ* mutant cells grown in a CMS-L medium and expressing recombinant proteins were harvested in a mid-exponential phase of growth (T0) and after 8-, 24-, and 48 h from T0. (**A,C**) Total protein extracts were prepared at indicated times, and samples were analyzed as described in Fig 1B. Anti-V5 and anti-HA antibodies were used to visualize Atg32-V5 or HA-Atg32 recombinant proteins. (**B,D**) The Atg32-V5/Pgk1 or HA-Atg32/Pgk1 ratios were quantified by using ImageJ software; *$P < 0.05$, **$P < 0.01$.

amount after 24 hours (S2A and S2B Fig). Moreover, Atg32-V5 degradation was not compromised under normal growth conditions in cells lacking Atg5 or Atg8, two essential autophagy effectors, or Atg11, a protein required for selective autophagy degradation (S3A and S3B Fig). Okamoto et al. (2009) described a similar disappearance of the Atg32 protein in *atg7Δ* mutant strains [4]. Assuming Atg32 is essential for only mitophagy, this finding was surprising. The reason for this finding is uncertain; however, a small quantity of Atg32 might be sufficient to mediate the selective elimination of mitochondria. Alternatively, a yet to be identified post-translational modification of Atg32 might function in mitophagy. Also, we cannot exclude the possibility that during respiratory growth, Atg32 might be involved in unknown pathways related to mitochondrial function.

## Atg32 turnover can be prevented by inhibition of the proteasome

To understand why the Atg32 protein level decreases when mitophagy is induced, we examined both possibilities that could lead to its reduction: increased degradation and a reduced synthesis of the Atg32 protein. As first, we evaluated the effect of inhibitors of the two main cellular proteolytic pathways (vacuolar-autophagy systems and the ubiquitin-proteasome) on the Atg32 protein level. We observed that the addition of PMSF, an inhibitor of serine proteases that blocks several yeast vacuolar proteases (e.g., proteinase B and carboxypeptidase Y) but does not affect proteasome function, only slightly affected Atg32 protein levels, in an

increment probably corresponding to the part of a protein that is degraded by mitophagy (Fig 3A and 3B; 24 h and 48 h). Accordingly, the degradation of Atg32-V5 was not compromised in cells lacking Pep4, a vacuolar protease (Fig 3C and 3D), confirming that degradation of Atg32 in the vacuole does not play a significant role in degradation control of the Atg32 protein.

Meanwhile, the addition of MG-132, a proteasome inhibitor that reduces the degradation of ubiquitin-conjugated proteins in mammalian cells and effectively blocks the proteolytic activity of the 26S proteasome in yeast, counteracted the Atg32 protein loss observed during the stationary growth phase (Fig 3A, upper panel) and rapamycin treatment (S4A Fig). We observed the same Atg32 protein level dynamic in the BY4742 strain expressing Atg32-V5 (S4B and S4C Fig). Further, inhibition of proteasomal activity stabilized the Atg32 protein in exponentially growing cells, in which there is no detectable mitophagy yet (T8; S4D Fig). However, MG-132 had only a limited effect on Atg32-V5 levels during nitrogen starvation (S2A

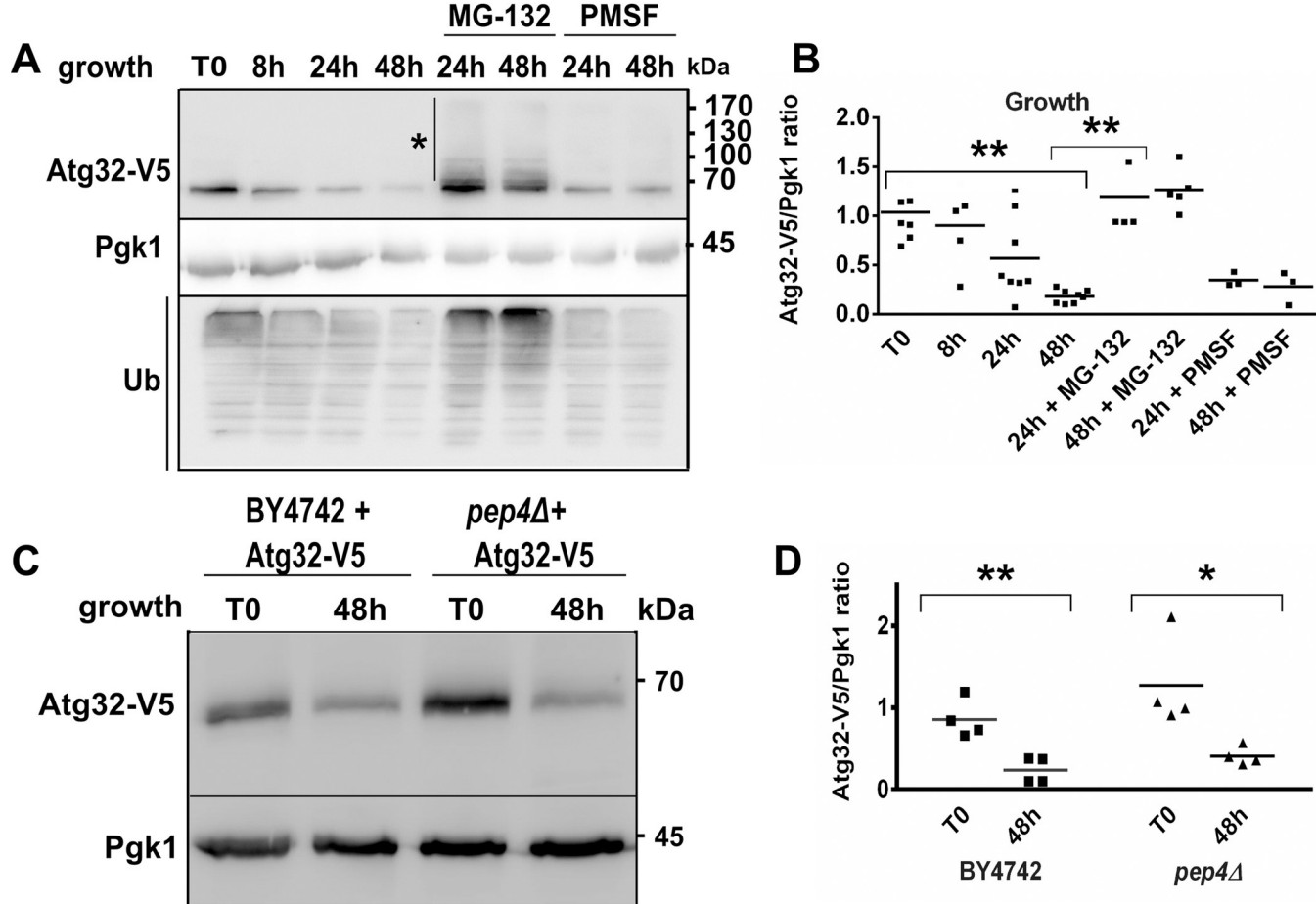

**Fig 3. Degradation of the Atg32 protein is prevented by inhibition of the proteasome. (A)** *atg32Δ* mutant cells grown in a CMS-L medium and expressing Atg32-V5 recombinant protein were harvested at indicated time points. To inhibit the proteasome, 75 μM MG-132 was added to the cell culture at the 8 h time point. To inhibit vacuolar proteolysis, 2 mM PMSF was added to the cell culture at the 8 h time point; this step was repeated several times during the whole course of cell growth. Total protein extracts were prepared, and protein samples were analyzed as described in Fig 1B. Anti-V5 antibody was used to visualize Atg32-V5 protein. Anti-ubiquitin (Ub) was used to detect the level of ubiquitinated proteins. **(B)** Atg32-V5 expression was quantified as the Atg32-V5/Pgk1 ratio; ** $P < 0.01$. **(C)** BY4742 and *pep4Δ* mutant cells expressing Atg32-V5 were grown in a CMS-L medium. Cells were harvested in a mid-exponential (T0) and a late stationary phase (48 h) of growth. Total protein extracts were prepared and analyzed by western blots. Anti-V5 antibody was used to visualize Atg32-V5 protein. **(D)** The Atg32-V5/Pgk1 ratio was quantified at T0 and 48 h time points for all tested strains—*$P < 0.05$ and **$P < 0.01$.

and S2B Fig). In Figs 3A and S2A (lower panels), the accumulation of ubiquitin-conjugated complexes (Ub) becomes easily detectable because these complexes are not degraded in the presence of MG-132; this serves as evidence that MG-132 is a potent proteasome inhibitor in yeast. Also, the Atg32-V5 level was restored with MG-132 treatment in *atg5Δ* mutant cells in the stationary phase (S3C Fig).

The addition of MG-132 or PMSF did not significantly affect cell growth or growth yield (S5A Fig). Furthermore, the simultaneous addition of MG-132 and PMSF had no significant cumulative effect on the reduction of Atg32 protein levels (S5B Fig). Consistent with these observations, Atg32 may be degraded in both an autophagy-dependent and autophagy-independent manner.

To answer the question of whether the reduction of the Atg32 protein level during the stationary phase in respiratory conditions is due to a decrease in Atg32 synthesis or an increase in Atg32 degradation, we examined *ATG32* promoter activity. We used a construct pPROM-*ATG32-lac Z* in which the reporter gene lacZ was expressed from the *ATG32* promoter. Interestingly, we found that *ATG32* promoter activity increased through the course of cell growth (Fig 4A). In the late stationary phase, *ATG32* promoter activity increased threefold to fourfold compared to *ATG32* promoter activity in cells in the early exponential phase of growth (T0). MG-132 treatment prevented this rise in *ATG32* promoter activity (Fig 4A).

To evaluate further whether the Atg32 protein is subject to degradation control, we monitored the Atg32 protein turnover. To assess the alteration of Atg32 protein levels, *atg32Δ* mutant cells expressing ATG32-V5 were grown in a minimal lactate medium, and in the mid-exponential growth treated with cycloheximide (250 μg/ml CHX) to turn off protein expression. After the first 20 minutes of CHX treatment, the Atg32-V5 protein level had already decreased dramatically compared to the levels of the cytosolic protein Pgk1 or of mitochondrial protein porin, indicating that Atg32 turnover is rapid (Fig 4B and 4C). Additionally, Atg32-V5 was significantly stabilized upon treatment by the proteasome inhibitor MG-132; hourly inhibition of the proteasomal turnover in combination with the cycloheximide resulted in a more-than-fourfold increase in the amount of the Atg32 protein (from 15 percent to approximately 68 percent; P = 0.015) (Fig 4B and 4C). These results suggest that the half-life of the Atg32 protein is very short and is degraded by the proteasome.

Consistent with the requirement of the chymotryptic activity of the proteasome in proteasome-mediated proteolysis, Atg32-V5 turnover was impaired in yeast mutant cells that contain a mutation in the protein Pre2 (Fig 5). Pre2 is the β5 subunit of the 20S proteasome, which is required for efficient assembly of the 20S proteasome catabolic core particle essential for degrading the protein into fragments [16]. The expression of the fusion protein Atg32-V5 decreased significantly in the stationary phase in the medium with galactose in the control strain (*atg32Δ* + Atg32-V5), just as it would in a lactate-containing medium, whereas we observe a less pronounced change for the *pre2-2* mutant strain (Fig 5). This supports the hypothesis of the involvement of the proteasome in the disappearance of the Atg32-V5 protein in the stationary phase of cell growth.

Our data showed that under respiratory conditions, the Atg32 protein had a high turnover rate. Its level decreased during mitophagy, whereas *ATG32* promoter activity increased suggesting the decrease of the Atg32 protein level during the stationary phase is due to an increase of Atg32 degradation. Our observations support the hypothesis that Atg32 might regulate its own protein level and activity.

The fact that treatment with MG-132 prevented the disappearance of Atg32 indicates that Atg32 protein degradation seems to be directly or indirectly dependent on proteasome activity. These results imply that Atg32 is ubiquitinated and targeted for proteasomal degradation. Accordingly, several bands exhibiting a molecular weight shift of Atg32-V5 (Fig 3A upper

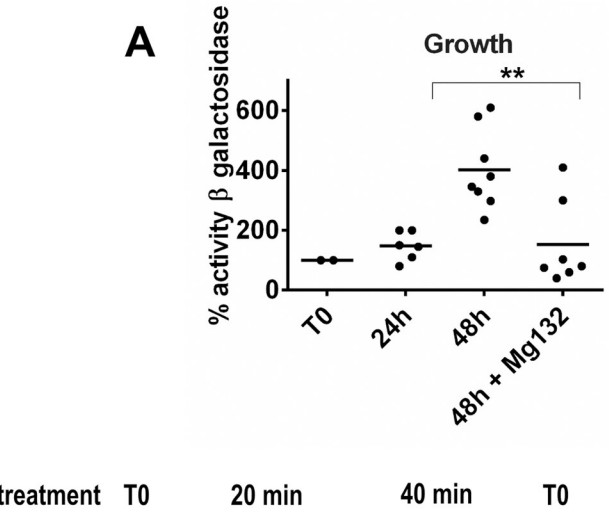

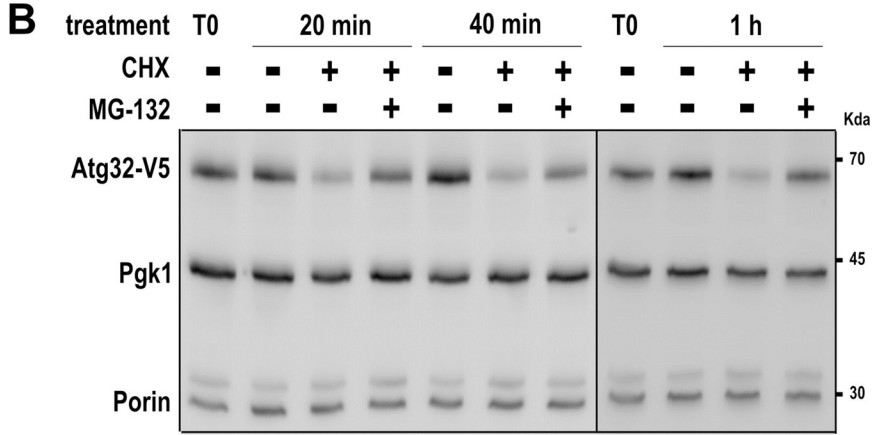

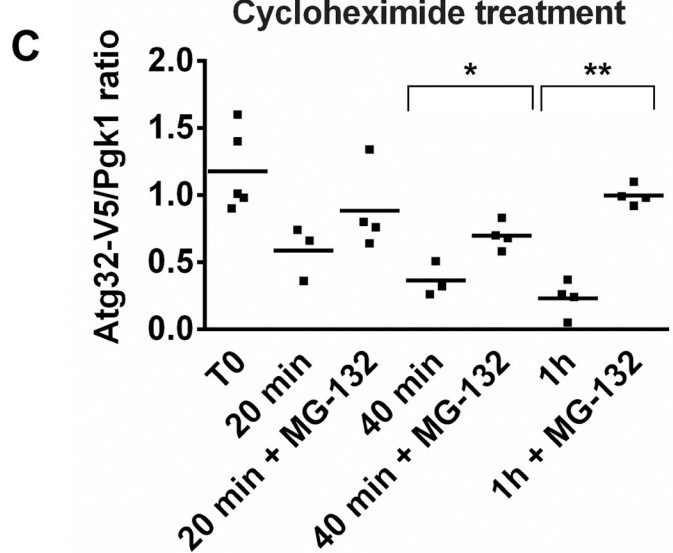

**Fig 4. Evaluation of *ATG32* promoter activity and Atg32 protein stability. (A)** BY4742 cells grown in a CMS-L medium and expressing *β*-galactosidase under the *ATG32* promoter were harvested in a mid-exponential phase of growth (T0) and after 24 h and 48 h from T0 in the absence or presence of MG-132. After cell lysis, the *β*-galactosidase activity was measured. Results were obtained from 6 independent experiments and are expressed as the % of T0. The symbol ** indicates significant difference between 48 h in the absence of and 48 h in the presence of MG-132 (*P*

<0.01). **(B)** *atg32Δ* mutant cells grown in a CMS-L medium and expressing Atg32-V5 protein were harvested in a mid-exponential phase (T0) and then treated with 250 μg/ml of cycloheximide (CHX) in the presence or absence of 75 μM MG-132 for 20 min, 40 min, and 1 h. Total protein extracts were analyzed by western blots. Anti-V5 antibody was used to visualize Atg32-V5 protein. Pgk1 was used as a loading control; Porin was used as a mitochondrial protein marker. At least 3 independent experiments were carried out for each condition. **(C)** Atg32-V5 expression was quantified as the Atg32-V5/Pgk1 ratio; * $P < 0.05$ and ** $P < 0.01$.

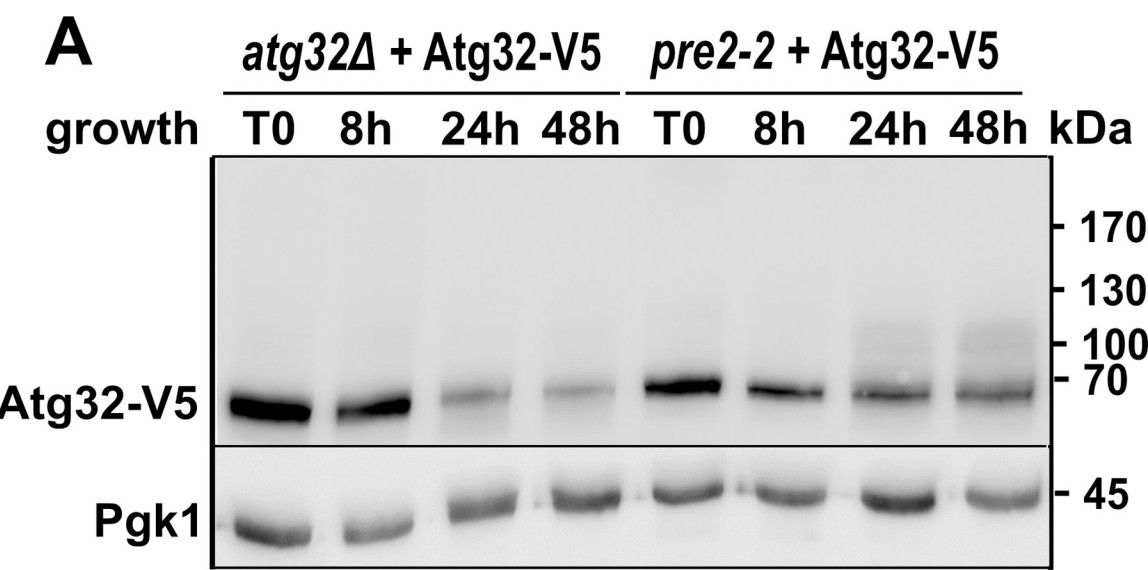

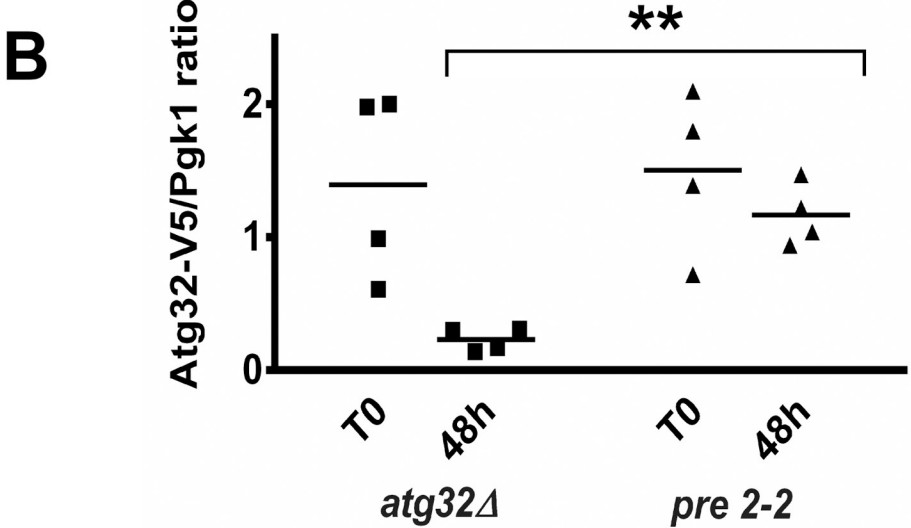

**Fig 5. Degradation of Atg32 protein is impaired in *pre2-2* mutant cells. (A)** atg32Δ and *pre2-2* mutant cells transformed with a plasmid expressing Atg32-V5 were grown in a CMS-G medium. Cells were harvested in a mid-exponential phase of growth (T0) and after 8, 24, and 48 h of cell growth. **(B)** The Atg32-V5/Pgk1 ratio was quantified at T0 and 48 h time points for all tested strains—** $P < 0.01$.

panel, asterisk mark; Figs 5A and S3C and S4A and S4B and S5B) ranging between 70 and 190 kDa were identified in protein extracts prepared from cells following MG-132 treatment or in *pre2-2* mutant cells. They may correspond to ubiquitinated Atg32 protein forms that were stabilized within cells due to dysfunctional proteasome.

## Inhibition of the proteasome with MG-132 stimulates mitophagy

It has been shown that cells lacking the Atg32 protein are deficient in mitophagy induction and that Atg32 overexpression is responsible for an increase in mitophagy [3, 4]. We showed Atg32-V5 is degraded under normal cell growth conditions (Figs 2, 3 and 5 and S4B), but the protein did not disappear in the stationary phase when cells were treated with MG-132 (Figs 2 and 3 and S4B) or in the *pre2-2* mutant (Fig 5). To check whether the higher Atg32 protein level in the stationary phase after MG-132 treatment correlated with an increase of mitophagy, we assessed mitophagy induction in cells treated or untreated with MG-132. We first studied mitophagy in BY4742 + Atg32-V5 or *atg32Δ* + Atg32-V5 cells growing in respiratory conditions and expressing the mitochondrial Idp1-GFP fusion protein. Under basal conditions, Idp1-GFP is exclusively located in the mitochondrial matrix, and only the Idp1-GFP band (Fig 6A and 6B; T0, 75 kDa) can be visualized by immunoanalysis. When mitophagy is induced, mitochondria are trapped within autophagosomes and delivered into the vacuole to be degraded. GFP moiety is much more resistant to hydrolase degradation than Idp1 moiety, and thus residual GFP (27 kDa) will remain in the vacuolar lumen and serves as evidence of mitophagy induction. In untreated control cells, the GFP band first appeared in the mid-exponential phase (8 h), indicating mitophagy was activated (Fig 6A and 6B). MG-132 stimulated both Idp1-GFP cleavage (Fig 6A–6C) and ATG32-V5 level (Fig 3), consistent with the published data [3, 4] demonstrating a direct correlation between Atg32 content and mitophagy activity. To confirm these results, we measured alkaline phosphatase (ALP) activity in BY4742 cells expressing the mitochondria-targeted Pho8Δ60 (mtPho8Δ60) protein [15]. In control cells, we observed a twofold increase in ALP activity in the late stationary phase (48 h) compared to the mid-exponential phase (T0; Fig 6D). Furthermore, at the 48 h point, MG-132 treatment induced a fourfold increase in ALP activity compared with that of T0, confirming proteasome inhibition by MG-132 causes an increase in mitophagy in the stationary phase compared to mitophagy of untreated cells. Notably, MG-132 had no stimulating effect on macroautophagy (S6 Fig).

## Identification of ubiquitination site in Atg32 protein by LC-MS/MS analysis

To confirm our hypothesis that the level and the activity of the Atg32 protein is regulated by ubiquitination, we purified Atg32 protein from *atg32Δ* mutant cells expressing Atg32-V5-6xHIS harvested in the late stationary growth phase (48 h) in the presence of MG-132. The cell lysate was loaded on an affinity Ni-NTA column, as described in the Material and Methods section. After elution, individual fractions were examined by western blot (S7 Fig) first. The presence of Atg32-V5 was confirmed by immunodetection with antibodies directed against histidine and ubiquitin (S7C and S7D Fig). Subsequently, fractions absorbing at 254 nm (S7A Fig) were pooled together, separated on 11% SDS-PAGE, and submitted to blue colloidal staining (Fig 7A) or immunoanalysis using antibodies directed against histidine and ubiquitin (Fig 7B), respectively.

Two bands (B1 and B2) were detected by colloidal blue staining (Fig 7A). Nevertheless, Atg32-V5 did not migrate at its expected mass of 58.9 kD. Although we detected a small amount of protein of theoretical size in eluted fractions (S7C Fig), most of the signal was

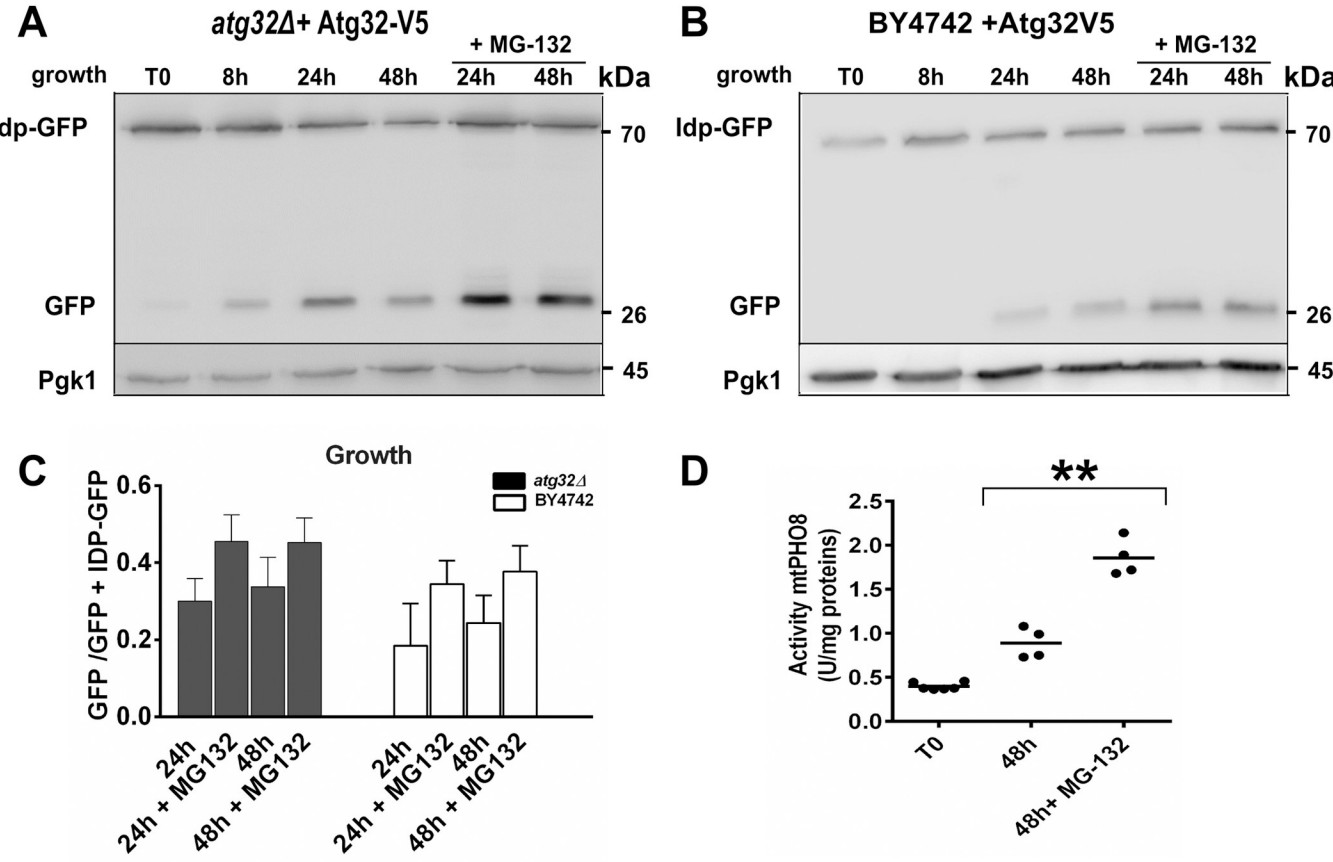

**Fig 6. Stabilization of Atg32 protein by proteasome inhibition stimulates mitophagy. (A,B)** *atg32Δ* mutant **(A)** and BY4742 **(B)** cells expressing Atg32-V5 and Idp1-GFP proteins grown in a CMS-L medium were harvested at indicated times of growth. To inhibit the proteasome, 75 μM MG-132 was added to the cell culture at the 8 h time point. Total protein extracts were prepared and analyzed as described in Fig 1B. Proteins were detected using antibodies against GFP or Pgk1. **(C)** GFP/GFP+Idp1-GFP ratios were quantified at the 24 h and 48 h time points for all tested conditions. Error bars represent standard error of the mean. **(D)** BY4742 cells expressing mtPHO8Δ60 protein grown in a CMS-L medium were harvested at the mid-exponential (T0) and late stationary (48 h) phases. MG-132 was added as described in (A). ALP activity was measured as described in the Material and Methods section. The symbol ** indicates significant difference between the 48-h time point with and without MG-132 ($P < 0.01$).

discovered in oligomer form with a high mass (larger than 170 kD) despite the presence of SDS. The shift in mobility of Atg32 after purification is a surprising finding but not impossible to understand. Atg32 is a protein inserted into the outer membrane of the mitochondria. It has already been observed that a mitochondrial membrane protein such as the ATP synthase subunit 9 is mainly found in the form oligomers in SDS-PAGE gel despite the presence of detergents in the gel [17, 18].

To ascertain Atg32 ubiquitination, the B1 and B2 bands were submitted to a standard proteomics workflow as described in the Material and Methods section, including an on-gel proteolysis step and an analysis by LC-MS/MS of peptides. Specifically, trypsin digestion of ubiquitinated proteins cleaves off all but the two C-terminal glycine residues of ubiquitin from the modified protein. These two C-terminal glycine (GG) residues remain linked to the epsilon amino group of the modified lysine residue in the tryptic peptide derived from the digestion of the substrate protein. The presence of the GG on the side chain of that Lys prevents cleavage by trypsin at that site, resulting in an internal modified Lys residue in a formerly ubiquitinated peptide.

Atg32 was unambiguously identified in a sample obtained from the B1 band based on 21 unique peptides leading to 55.6% sequence coverage (Table 1 and S8 Fig). One of these

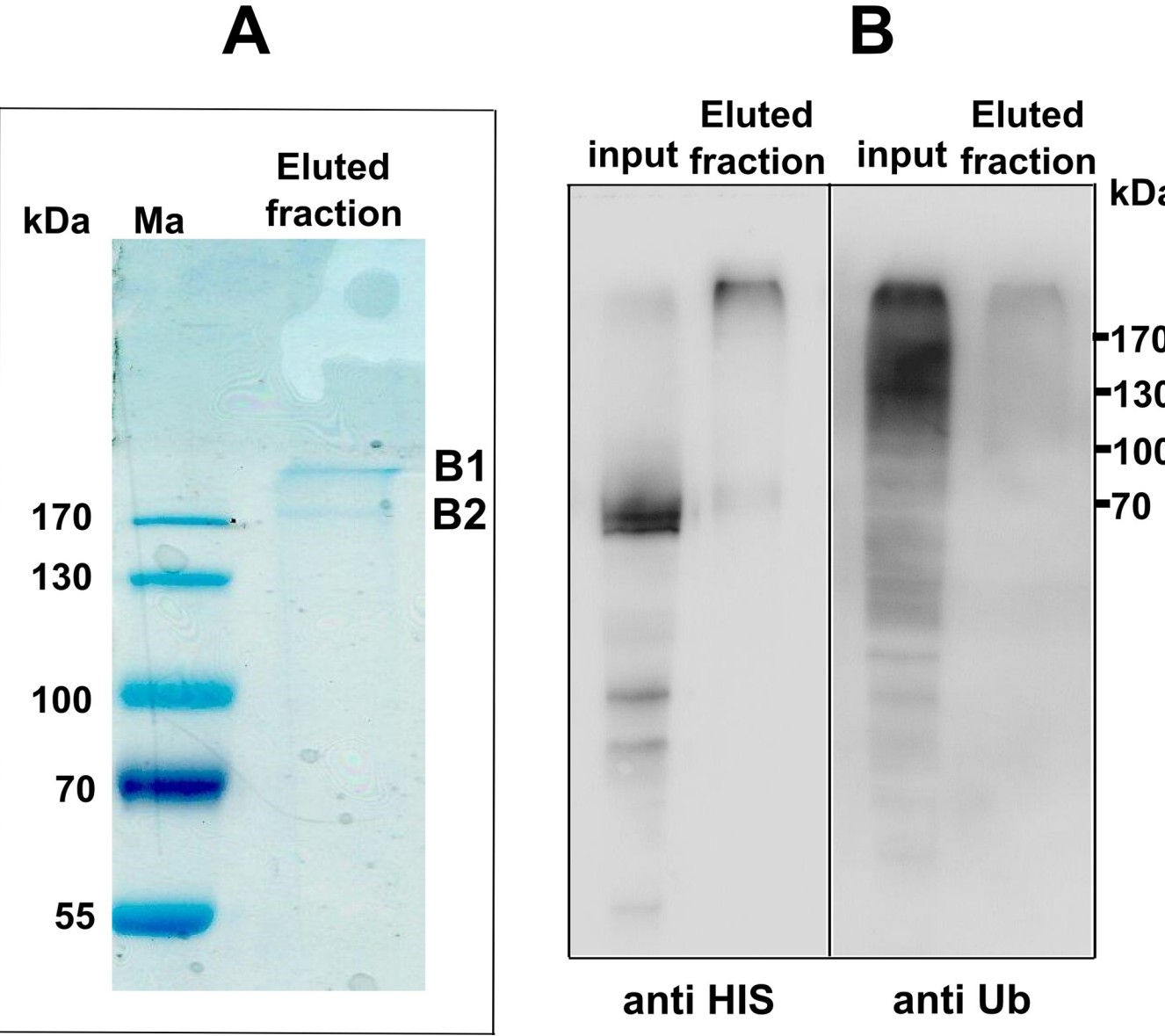

**Fig 7. Atg32 purification and mass spectrometry analysis. (A)** *atg32Δ* mutant cells expressing Atg32-V5-6HIS were grown in a CMS-L medium until a late stationary phase (48 h) in presence of MG-132. Cells were lysed and loaded on a Ni-NTA column as described in the Material and Methods section. After the last elution, collected fractions absorbing at 254 nm (from 14 to 17) were pooled together, precipitated with TCA. Pellet was resuspended in 60 μl of the loading buffer, 50 μl (eluted fraction) were loaded on a 11% SDS-PAGE. After migration, the gel was colored with colloidal blue. The bands B1 and B2 were excised and analyzed by mass spectrometry. **(B)** 5 μL of pooled fraction from A (Eluted fraction) were separated in a 11% SDS-PAGE gel. Proteins were analyzed by western blots using antibodies against histidine (anti HIS) or ubiquitin (anti Ub), respectively. Input: 50 μL of cell lysate prepared in (A) was precipitated with TCA, pellet was resuspended in 50 μL of the loading buffer, and 4 μL were loaded on the gel.

peptides was unambiguously shown to bear the typical GG tag (+144 Da) remaining after tryptic proteolysis of ubiquitination of Lysines using two distinct search engines (SEQUEST and MASCOT) and manual validation (visual inspection of the related MS/MS spectrum). The simultaneous detection into the MS/MS spectrum of 20 b fragments and 22 y fragments (over 22 possible fragments of each) ascertains the presence of a typical GG tag on the lysine 282 and confirms at least one ubiquitination site into Atg32 (Table 1 and S8 Fig). However, the Atg32 protein contains 43 lysines, and only 17 were covered by MS analysis. Thus, 26 lysines remain

**Table 1. List of peptides found after mass spectrometry analysis of Atg32 protein.**

| Annotated sequence | Modifications |
| --- | --- |
| [K]. GTLQINFHSDGFIMK. [S] | 1xOxidation [M14] |
| [K]. SLTSSTNSFVMPK.[L] | 1xOxidation [M14] |
| [K]. SMPPDSSSTTIHTCSEAQTGEDK.[G] | 1xCarbamidomethil [C14]; 1xOxidation [M2] |
| [R]. IVQYSQGKPVIPICQPGQVIQVK.[N] | 1xCarbamidomethil [C14]; 1xGG [K8] |
| [R]. IVQYSQGKPVIPICQPGQVIQVK.[N] | 1xCarbamidomethil [C14] |
| [K]. SMPPDSSSTTIHTCSEAQTGEDK.[G] | 1xCarbamidomethil [C14] |
| [K]. SMPPDSSSTTIHTCSEAQTGEDKGLLDPHLSVLELLSK.[T] | 1xCarbamidomethil [C14] |
| [R]. CSSQTTNGSILSSSDTSEEEQELLQAPAADIINIIK.[Q] | 1xCarbamidomethil [C1] |
| [K]. MNTFVLHALSKPLK.[F] | |
| [K]. QGQEGANVVSPSHPFK.[Q] | |
| [M]. VLEYQQR.[E] | |
| [K]. YHDSATFPQYTGIVIIFQELR.[E] | |
| [K]. SLTSSTNSFVMPK.[L] | |
| [K]. SSEFSIDESNR.[I] | |
| [K]. TPFENQDDDGDEDEAFEEDSVTITK.[S] | |
| [R]. TGSSFYQSIPK.[E] | |
| [K]. EYQSLFELPK.[Y] | |
| [K]. EKTPFNEQDDDGDEDEAFEEDSVTITK.[S] | |
| [R]. EMVSLLNR.[I] | |
| [K]. FLENLNK.[S] | |
| [K]. LLFPPVVVTNK.[R] | |
| [K]. LLFPPVVVTNKR.[D] | |
| [R]. LQDLSLEYGEDVNEEDNDDEAIHTK.[S] | |
| [K]. GLLDPHLSVLELLSK.[T] | |
| [K].GTLQINFHSDGFIMK.[S] | |

to be tested. In a sample obtained from the band B2, MS analysis identified a histidine-rich protein which was also found ubiquitinated.

## Investigation of Atg32 expression in ubiquitination-deficient Atg32 mutants

To understand the specific role of proteasome-mediated Atg32 turnover, the identification of the specific pathway that selects Atg32 for degradation is required.

Because our principal aim was to determine the involvement of lysine 282 residue in Atg32 turnover, we replaced lysine 282 with alanine (mutant K282A). We found that the Atg32$^{K282A}$-V5 mutant form is also degraded during growth. When compared to wild-type Atg32 protein, there is about 10–15% ($P<0.05$) more mutant protein in the late stationary phase cells (Fig 8A and 8C; 48 h). The fact that Atg32 expression was not fully restored in the K282A mutant could indicate that additional lysine residues on Atg32 might also be ubiquitinated and thus involved in degradation control and regulation of mitophagy activity of Atg32.

When the Atg32 protein sequence was analyzed, we noticed the presence of an Rsp5-binding motif (Leu-Pro-Lys-Tyr, LPKY) at the position 243–246. The *RSP5* gene encodes an essential ubiquitin ligase. Proteasomal substrates are usually first recognized by a ubiquitin ligase (E3), which works with a ubiquitin-activating enzyme (E1) and a ubiquitin-conjugating enzyme (E2) to decorate the substrates with the ubiquitin molecule. Ubiquitin-modified substrates are then delivered to the proteasome for degradation [19]. Because the ubiquitin ligase E3 is the rate-

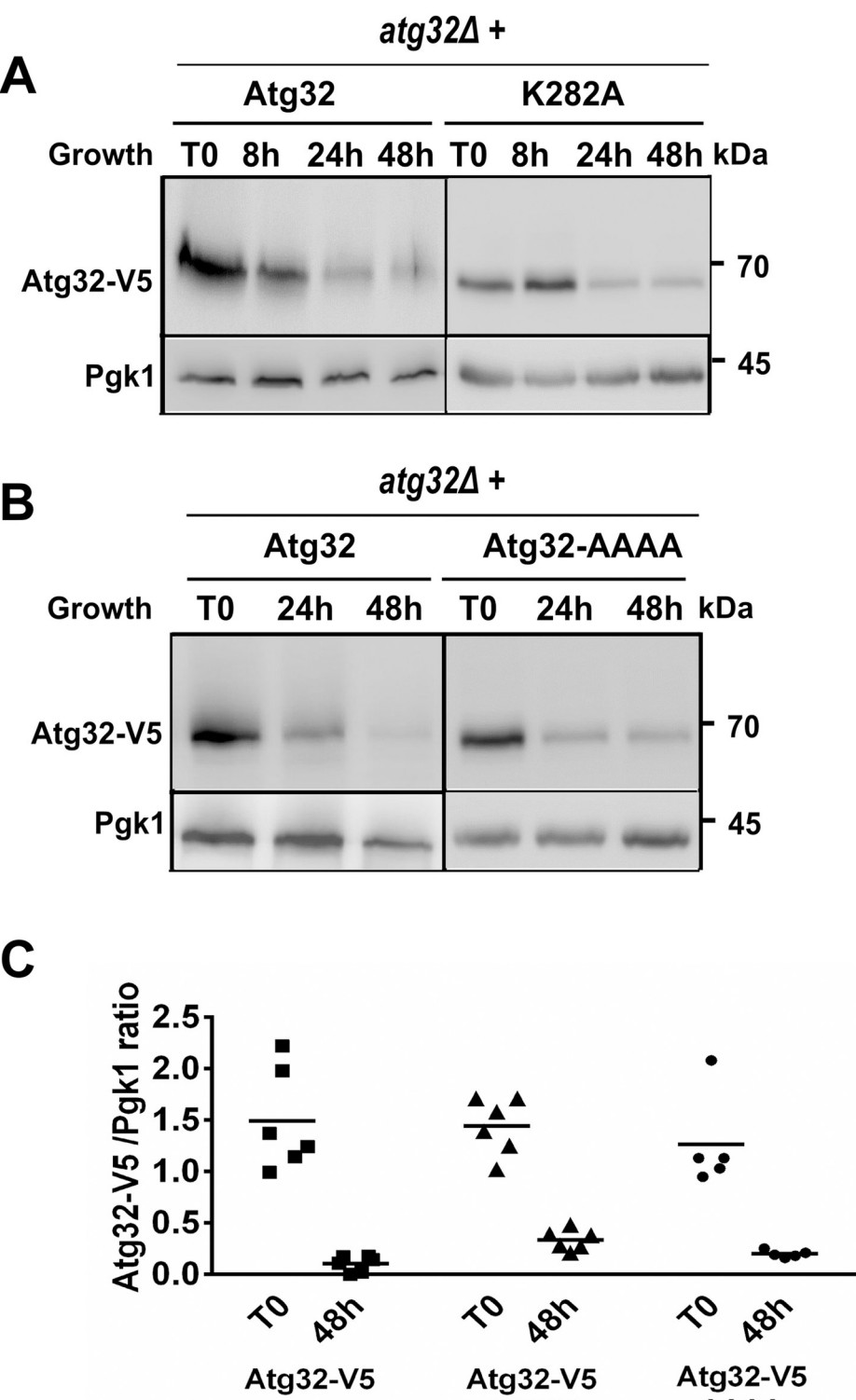

**Fig 8. Evaluation of Atg32 degradation in ubiquitination-deficient Atg32 mutants.** *atg32Δ* cells grown in a CMS-L medium and expressing wild-type Atg32-V5 or Atg32-V5[K282A] (**A**) or Atg32-V5 with mutated *RSP5*-binding motif (Atg32-AAAA-V5) (**B**) recombinant proteins were harvested at indicated times. Total protein extracts were prepared, and samples were analyzed by western blots. Anti-V5 antibody was used to visualize Atg32-V5 protein. (**C**) Wild-type or mutant forms of Atg32-V5 protein levels were quantified as the Atg32-V5/Pgk1 ratio. The symbol * indicates

significant difference between Atg32-V5 and Atg32$^{K282A}$-V5 on one side and Atg32-V5 and Atg32-AAAA-V5 on the other side ($P < 0.05$).

limiting and substrate-recognition component of the proteasome system, we examined the involvement of Rsp5 in Atg32 turnover. We replaced the LPKY motif in the *ATG32* sequence with AAAA (Ala-Ala-Ala-Ala), and we evaluated the expression of this mutant protein during cell growth. The turnover rate of the Atg32-AAAA-V5 protein was significantly ($P<0.05$) impaired compared to the wild-type protein (Fig 8B and 8C). This result thus indicates that the ubiquitination of Atg32 is not a straightforward process and could be (i) executed at various other lysine residues besides lysine 282 and (ii) governed by several E3 ligases (from more than 60 identified) involved in ubiquitin-mediated degradation in the yeast *S. cerevisiae*.

Next, we assessed mitophagy induction in the stationary phase in the different mutant cells potentially involved in Atg32 ubiquitination, such as Atg32$^{K282A}$ and Atg32-AAAA. We observed that the mitophagy activity in these two mutants is very similar to that of the control strain, with only a slight but significant increase in strains expressing K282A ($P<0.05$) or Atg32-AAAA ($P<0.01$) (Fig 9A and 9B).

These data suggest the link between Atg32 expression, its turnover, and that mitophagy may be more complex than we expected.

## Discussion

### Regulation of mitophagy by ubiquitination

Atg32 executes a key role as a receptor for multiple mitophagy inducing pathways and governs the final turnover of mitochondria. Therefore, a thorough regulation of its activity is expected. Besides regulation of its gene expression, post-translational modifications seem to control Atg32-mediated mitophagy. However, the pathways perceiving mitochondrial damage/impairment and Atg32 activation are not fully understood.

In this work, we were interested in (i) the expression and the stability of the Atg32 protein during mitophagy and (ii) the relationship between mitophagy, Atg32, and the proteasome. We observed that the Atg32-V5 protein was expressed in the mid-exponential phase of growth, disappeared progressively with growth, and was almost entirely missing in the stationary phase of growth as well as during longer durations of nitrogen starvation or upon rapamycin treatment. Several groups have already shown that Atg32 protein was expressed during growth in respiratory conditions and disappeared in the stationary phase of growth [4, 20]. Wang et al. (2013) demonstrated that Atg32 is processed by Yme1 protease, resulting in Atg32 C-terminus cleavage, and this processed form is required for mitophagy [20]. We tagged Atg32 protein with a V5 tag in the C-terminus, and this recombinant protein may be subjected to Yme1 dependent processing, resulting in the loss of the V5 tag upon mitophagy induction. However, in Wang *et al.*'s study, Atg32 processing by Yme1 was weak and slow in cells grown in the presence of lactate as a carbon source, only about 5% of Atg32 was cleaved by Yme1 in both nitrogen starvation and in the stationary phase of growth [20]. Moreover, in the same article, the authors showed the disappearance of Atg32 tagged in N-terminus with a TAP tag in the stationary phase of growth, and we also observed Atg32 turnover with the use of HA-Atg32 fusion protein. Therefore, our results concerning the disappearance of Atg32-V5 in the stationary phase of growth are in line with the published data, and Yme1 probably only plays a minor role in the disappearance of Atg32-V5 recombinant protein in this phase.

Considering the important role of the proteasome in the regulation of numerous proteins of the outer mitochondrial membrane [21, 22], we assessed whether the proteasome is also

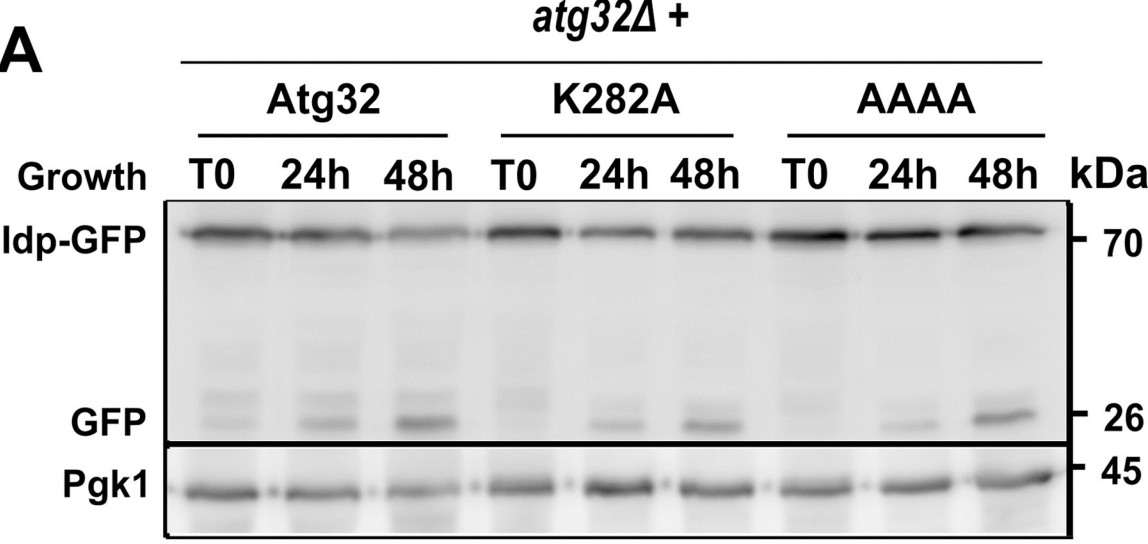

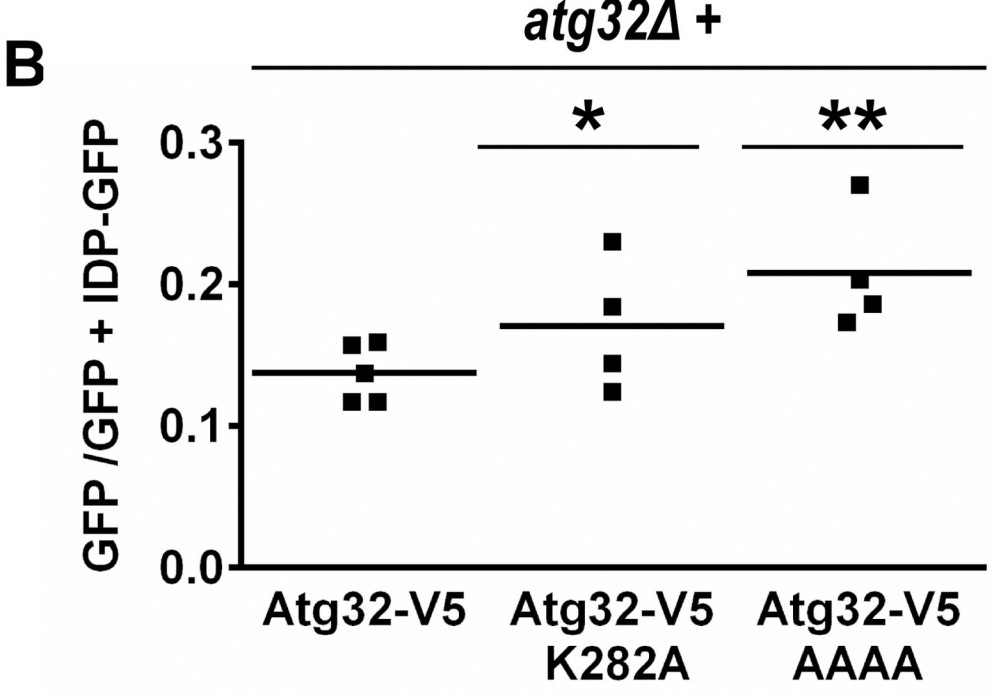

**Fig 9. Evaluation of mitophagy in ubiquitination-deficient Atg32 mutants. (A)** *atg32Δ* cells grown in a CMS-L medium and expressing Idp1-GFP fusion protein and Atg32-V5 recombinant protein with mutation K282A or mutated Rsp5-binding motif (AAAA) were harvested at indicated times. Mitophagy was assessed using mitophagy-dependent processing of the Idp1-GFP tool as described in Fig 1B. **(B)** Mitophagy induction was quantified as the ratio of GFP/(Idp1-GFP+GFP). The symbols ** and * indicate significant differences between Atg32-V5 and Atg32-AAAA-V5 ($P < 0.01$) or between Atg32-V5 and Atg32-K282A ($P < 0.05$), respectively.

involved in Atg32 degradation. Indeed, blocking the proteolytic activity of proteasome by MG-132 allowed us to prevent a decline in the Atg32 levels through the course of growth and in the stationary phase and to stabilize the modified form of Atg32. Moreover, in the proteasome *pre2-2* mutant strain, the Atg32 levels decreased to a much lesser extent than did control cells. Furthermore, the limited effect of treatment with PMSF, an inhibitor of vacuolar serine proteases, or of the deletion of the vacuolar protease Pep4 indicated that only a small part of Atg32 is degraded by mitophagy. We showed that the half-life of the Atg32 protein is short and that Atg32 protein levels are regulated by proteasome activity. Moreover, our results suggest that this regulatory mechanism is already active in exponentially growing cells, hence, before mitophagy induction. To some extent, our results differ from the observations that Levchenko et al. (2016) published. Besides differences in strains, the only explanation we can offer is that a difference in growing conditions and/or mitophagy induction leads to the involvement of various pathways regulating expression/stability of Atg32; one pathway can be more dependent on vacuolar degradation while another can undergo proteasomal degradation control.

Previous studies have shown that overexpression of Atg32 stimulates mitophagy, while it has long been known that the absence of Atg32 protein causes mitophagy impairment in yeast [3, 4]. These published data suggest that mitophagy is dependent on the amount of Atg32 protein present. Our results are in line with this observation. We showed that upon proteasome inhibition with MG-132, there is an increase in the amount of Atg32 protein, which correlates with the stimulation of mitophagy in the stationary phase of growth when compared to control (untreated) cells. It is expected that during this stimulation/induction of mitophagy, the amount of Atg32 protein is crucial, and our findings suggest that the proteasome may regulate the Atg32 protein amount so that yeast cells can control mitophagy activity.

Recently, it was found that two other Atg proteins are regulated by ubiquitination: Atg14 is degraded by the ubiquitin-proteasome system as a means to modulate autophagy activity [23], and Atg9 is degraded by the proteasome under normal conditions but stabilized upon starvation and rapamycin treatment [24].

Altogether, our data indicate that, as in mammalian cells, an interplay exists between mitochondria degradation by autophagy and the proteasome. It has been shown that outer mitochondrial membrane ubiquitin ligases, such as mammalian MULAN, MARCHV/MITOL, and yeast Mdm30, can ubiquitinate proteins involved in mitochondrial fusion and fission, targeting them for proteasome degradation and thus affecting mitochondrial dynamics [25–27]. These ligases ensure outer mitochondrial membrane quality control. Moreover, the Rsp5 E3 ligase has been shown to ubiquitinate Mdm34 and Mdm12, two components of ERMES, and this event is required for efficient mitophagy in yeast [28].

In mammals, mitophagy is governed by two different pathways. Numerous data show that the UPS plays a major role in Parkin-dependent mitophagy. Chan et al. (2011) observed robust recruitment of the 26S proteasome onto mitochondria, leading to widespread degradation of mitochondrial outer membrane proteins via the UPS [29]. Strikingly, activation of the UPS not only precedes mitophagy but is also required for mitophagy. Inhibition of the UPS causes complete abrogation of mitophagy [27]. Tanaka *et al.* showed that ubiquitination of the mitofusins Mfn1 and Mfn2, two large GTPases that mediate mitochondrial fusion, is induced by Parkin upon membrane depolarization and leads to their degradation in a proteasome- and p97-dependent manner [30]. More recently, Wei *et al.* showed that proteasome-dependent mitochondrial membrane rupture is necessary for Parkin-mediated mitophagy in mammalian cells, in part via the cytoplasmic exposure to an IMM mitophagy receptor [31]. Concerning the pathway involving receptors, ubiquitination could play a different role. In mammalian cells, such a situation was observed in the case of the FUNDC1 receptor, which is primarily involved in mitophagy during hypoxia. This receptor is regulated by both phosphorylation

and ubiquitination [32–35]. The mitochondrial PGAM5 phosphatase interacts with and dephosphorylates FUNDC1 serine 13 (Ser-13) residue upon hypoxia or carbonyl cyanide p-tri-fluoromethoxyphenylhydrazone treatment. Dephosphorylation of FUNDC1 catalyzed by PGAM5 enhances its interaction with LC3. CK2 phosphorylates FUNDC1 to reverse the effect of PGAM5 in mitophagy activation. Indeed, the mitochondrial E3 ligase MARCH5 plays a role in regulating hypoxia-induced mitophagy by ubiquitinating and degrading FUNDC1 to limit excessive mitochondria degradation [34]. Our study shows that, in yeast, in addition to phosphorylation regulation, Atg32-V5 turnover seems to be dependent on proteasome activity, and lysine 282 and Rsp5-binding motif are identified as one of the targets of ubiquitination. The fact that the mutation of the Rsp5-binding motif, Rsp5 being an E3 ligase, slightly but significantly perturbs Atg32 expression and mitophagy could indicate that this E3 ligase is involved in the regulation of Atg32. However, we can not exclude the participation of other E3 ligases. Indeed, the effect of these mutations on mitophagy is rather subtle compared to the data obtained with inhibition of the proteasome by MG-132. This could be due to the presence of various different ubiquitination sites present in *ATG32* sequence.

Further study aimed at revealing other potential ubiquitination sites in Atg32 protein as well as examining strains with mutated both ubiquitination and phosphorylation sites could shed more light on this process.

Ubiquitination of mitochondrial proteins can play two different roles: (i) ubiquitination of proteins allows their turnover by proteasomal degradation and, consequently, their abundance and quality control; and (ii) ubiquitination of proteins localized on the outer mitochondrial membrane is a signal to trigger mitophagy. In the case of Atg32 or FUNDC1 mitophagy receptors, different post-translational modifications take place to modulate its activity and regulate mitophagy level. We can hypothesize that ubiquitination will be useful for regulating Atg32's own expression and controlling the mitophagy level. This would prevent excessive mitochondria degradation and ensure a fine-tuned mitochondria turnover.

## Why is Atg32 regulated by multiple posttranslational modifications?

During mitophagy induction, Atg32 is activated by a post-translational modification. This, however, does not answer the question of why the Atg32 protein is subjected to several different post-translational modifications, such as phosphorylations, ubiquitination, or other unknown modifications, and what the roles of these modifications are. Are these modifications dependent on mitophagy induction conditions (e.g., nutrient starvation, stationary phase of growth, or rapamycin treatment)? Previously, we showed that depending on the condition of induction, mitophagy may exploit various signaling pathways [36]. The need of cells to employ various pathways could reflect different purposes of self-eating mitochondria in cell physiology at certain moments, the bioenergetic state/deficit of cells, a cell's capacity to carry a burden of damaged components, and so on. Upon starvation and in the stationary phase, mitophagy requires the same key proteins, Atg32, Atg11, and Atg8; however, in this study we demonstrated that Atg32 turnover rates differ under both conditions, which supports the idea that Atg32 can be regulated through more than one mechanism. In mammalian cells, mitophagy also has several distinct variants based on the biological distinctions of cells: Type 1 mitophagy, which occurs during nutrient deprivation; Type 2 mitophagy, which is stimulated by mitochondrial damage; and micromitophagy [37].

From the published data, phosphorylation of Atg32 at serine 114 and serine 119 by a serine/threonine protein kinase Ck2 seems to be required for the interaction with the adaptor protein Atg11 and targeting mitochondria for a degradation into the vacuole [38]. Thus, vacuolar degradation is responsible for Atg32 stability bearing this kind of modification. What is the fate of

Atg32 on mitochondria that have not been selected for mitophagy? We assumed that Atg32 could be eliminated by a separate pathway of degradation in its unmodified form or after acquiring another type of modification. In fact, our results suggest that Atg32 could also be ubiquitylated, and mitophagy can be regulated by ubiquitination/deubiquitination events.

Additional experiments are needed to further investigate how cells control the level and activity of Atg32 to be able to understand the mechanisms by which cells control selective degradation of mitochondria, and the physiological significance of mitophagy. In addition, an interesting question is whether Atg32 is also involved in other cellular processes. Future studies of yeast will address these important questions.

## Supporting information

**S1 Fig. Atg32-V5 recombinant protein localizes into mitochondria and restores mitophagy in** *atg32Δ* **mutant cells upon nitrogen starvation. (A)** *atg32Δ* mutant cells grown in a CMS-L medium and expressing Atg32-V5 protein were harvested in a mid-exponential phase of growth. Then, cells were lysed and purified mitochondria prepared as described in Vigie et al. [34] were loaded on a 20–60% sucrose gradient. Fractions were collected and analyzed by immunodetection. Anti-V5 antibody was used to visualize Atg32-V5, and anti-Porin antibody was used to detect mitochondria-containing fractions. **(B)** Mitophagy was assessed using mitophagy-dependent processing of the Idp1-GFP tool in *atg32Δ* mutant and *atg32Δ*–expressing Atg32-V5. Cells were harvested at indicated times. The total protein extracts corresponding to $5 \times 10^6$ cells/50 ug proteins per line were separated by 12.5% SDS-PAGE and analyzed by immunodetection using anti-GFP antibody.
(PDF)

**S2 Fig. The Atg32 protein is degraded upon nitrogen starvation. (A)** *atg32Δ* mutant cells grown in a CMS-L medium and expressing Atg32-V5 protein were harvested at indicated times. To inhibit the proteasome, 75 μM MG-132 was added to the cell culture at the beginning of starvation. Total protein extracts were prepared afterwards, and samples were analyzed by western blots. Anti-V5 antibody was used to visualize Atg32-V5 protein; Pgk1 was used as a loading control. Anti-ubiquitin (Ub) was used to detect the level of ubiquitinated proteins. **(B)** The Atg32-V5/Pgk1 ratio was quantified for all tested conditions.
(PDF)

**S3 Fig. Degradation of Atg32 protein is not impaired in autophagy-deficient mutants under normal growth condition. (A)** *atg5Δ*, *atg8Δ*, and *atg11Δ* mutant cells transformed with a plasmid expressing Atg32-V5 were grown in a CMS-L medium. Cells were harvested at indicated times. **(B)** The Atg32-V5/Pgk1 ratios were quantified at T0 and 48 h time points for all tested strains; ** p<0.01. **(C)** *atg5Δ* mutant cells expressing Atg32-V5 were treated with MG-132 at time point 8 h. Cells were harvested at indicated time points and total protein extracts were prepared and analyzed by immunodetection. Anti-V5 antibody was used to visualize Atg32-V5 protein.
(PDF)

**S4 Fig.** (A) The Atg32 protein is degraded upon rapamycin treatment and stabilized by the proteasome inhibition. atg32Δ mutant cells grown in a CMS-L medium and expressing Atg32-V5 protein were harvested at T0 and treated with 0.2 μg/ml rapamycin in presence or absence of 75 μM MG-132 for 3 h, 6 h, and 24 h. Total protein extracts were prepared afterwards, and samples were analyzed by immunodetection. Anti-V5 antibody was used to visualize Atg32-V5 protein. (B) The Atg32 protein is degraded in BY4742 strain. BY4742 cells transformed with a plasmid expressing Atg32-V5 grown in a CMS-L medium were harvested

at indicated times. To inhibit proteasome, MG-132 was added to the cell culture at 8 h time point. (C) The Atg32-V5/Pgk1 ratios were quantify for all tested conditions—**P<0.01. (D) MG-123 stabilizes the Atg32 protein in exponentially growing cells. atg32Δ mutant cells grown in a CMS-L medium and expressing Atg32-V5 protein were harvested at T0 and treated with 75 μM MG-132. Cells were harvested at exponential (T8) and stationary (T24, T48) phase, and total protein extracts were prepared and analyzed by immunodetection. Anti-V5 antibody was used to visualize Atg32-V5 protein.
(PDF)

**S5 Fig. The effect of MG-132 and PMSF treatment on cell growth and Atg32-V5 protein degradation. (A)** Addition of proteasome inhibitor MG-132 (75 μM MG-132) and inhibitor of vacuolar proteolysis PMSF (2 mM) do not affect growth and growth yield in *atg32Δ* mutant cells expressing Atg32-V5 plasmid and grown in a CMS-L medium. The Y-axis is represented in logarithmic scale (n = 5 for control and MG-132; n = 3 for PMSF). **(B)** *atg32Δ* cells grown in a CMS-L medium and expressing Atg32-V5 protein were harvested at indicated time points. To inhibit proteasome, MG-132 was added to the cell culture at 8 h time point. To inhibit vacuolar proteolysis, 2 mM PMSF was added to the cell culture at T8; this step was repeated twice during the course of cell growth. Total protein extracts were prepared afterwards, and samples were analyzed by western blots. Anti-V5 antibody was used to visualize Atg32-V5 protein. To detect modified Atg32-V5 forms (bands with a higher molecular weight) after MG-132 treatment, two different revelation times of blots are presented.
(PDF)

**S6 Fig. Inhibition of the proteasome with MG-132 does not affect autophagy.** BY4742 **(A)** and *atg32Δ* mutant **(B)** cells expressing GFP-Atg8 protein grown in a CMS-L medium in presence or absence of MG-132 were harvested at indicated times. Total protein extracts from 2 x 10$^7$ cells were prepared and separated by 12.5% SDS-PAGE gel as described in the Material and Methods section. Proteins were detected using antibodies against GFP or Pgk1.
(PDF)

**S7 Fig. Purification of Atg32-V5-6HIS.** (A) Lysate from *atg32Δ* mutant cells expressing *pATG32-V5-6HIS* was prepared as described in the Material and Methods section. Next, lysate was loaded on a Ni-NTA column, the non-retained fraction, as well as the two washes W1 and W2, were recovered. The bounded proteins were then eluted and 500 μl fractions were collected. **(B-D)** 250 μl of each fraction absorbing at 254 nm (from F12 to F22) were precipitated with TCA. Pellets were resuspended in 20 μl of the loading buffer; 10 μl were loaded on the gel to be revealed with the colloidal blue (B) and 5 μl were used for immunodetection with anti-histidine (C) or anti-ubiquitin antibodies (D), respectively.
(PDF)

**S8 Fig. Mass spectrometry analysis.** (A) SEQUEST spectra, (B) MASCOT spectra, and (C) Protein coverage.
(PDF)

**S1 Table.**
(PDF)

**S1 File.**
(PDF)

**S2 File.**
(PDF)

## Acknowledgments

We thank Drs Sagot, Klionsky, Pinson and Abeliovich, and Mr Athané for the gift of strains and material and Drs Chaignepain and Claverol (CGFB, Bordeaux) for helpful discussions.

## Author Contributions

**Conceptualization:** Nadine Camougrand, Ingrid Bhatia-Kiššová.

**Data curation:** Nadine Camougrand, Pierre Vigié.

**Formal analysis:** Nadine Camougrand, Cécile Gonzalez.

**Methodology:** Nadine Camougrand, Cécile Gonzalez.

**Validation:** Nadine Camougrand.

**Writing – original draft:** Nadine Camougrand, Ingrid Bhatia-Kiššová.

**Writing – review & editing:** Stéphen Manon.

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
