## [Decision Letter · Decision Letter 0]

14 May 2020

PONE-D-20-11526

The yeast mitophagy receptor Atg32 is ubiquitinated and degraded by the proteasome

PLOS ONE

Dear Dr. Camougrand,

Thank you for submitting your manuscript to PLOS ONE. After careful consideration, we feel that it has merit but does not fully meet PLOS ONE’s publication criteria as it currently stands. 

Two expert reviewers have carefully examined your manuscript, and I have read the manuscript as well.

Certainly, the manuscript requires very considerable revision. Of note, the construct used has not been demonstrated to fully recapitulate the behavior of endogenous Atg32p, due to the presence of a carboxyl-terminal tag and potential over-expression from a plasmid. Mitochondrial localization of your construct has not been adequately demonstrated. It would be important to show that endogenous Atg32p, or a chromosome-integrated, internally tagged Atg32p previously demonstrated to perform as endogenous Atg32p, localizes to mitochondria and behaves similarly in key assays.

Both reviewers raise additional, substantial issues regarding whether the manuscript is technically sound, and these points should be addressed in a revised manuscript.

Finally, the results reported are in conflict with previous findings. In a revised manuscript, the authors should directly address and rationalize differences with earlier publications.

We would appreciate receiving your revised manuscript by Jun 28 2020 11:59PM. To enhance the reproducibility of your results, we recommend that if applicable you deposit your laboratory protocols in protocols.io, where a protocol can be assigned its own identifier (DOI) such that it can be cited independently in the future. For instructions see: http://journals.plos.org/plosone/s/submission-guidelines#loc-laboratory-protocols

We look forward to receiving your revised manuscript.

Kind regards,

Cory Dunn, Ph.D.

Academic Editor

PLOS ONE

3. Please include your tables as part of your main manuscript and remove the individual files. Please note that supplementary tables (should remain/ be uploaded) as separate "supporting information" files

Reviewers' comments:

Reviewer's Responses to Questions

**Comments to the Author**

1. Is the manuscript technically sound, and do the data support the conclusions?

Reviewer #1: No

Reviewer #2: No

2. Has the statistical analysis been performed appropriately and rigorously? 

Reviewer #1: Yes

Reviewer #2: No

3. Have the authors made all data underlying the findings in their manuscript fully available?

Reviewer #1: Yes

Reviewer #2: No

4. Is the manuscript presented in an intelligible fashion and written in standard English?

Reviewer #1: Yes

Reviewer #2: Yes

5. Review Comments to the Author

Reviewer #1: In this manuscript, Camougrand et al examine the regulation of the yeast selective autophagy receptor, Atg32. The authors observe that while Atg32 is specifically expressed during growth in non-fermentable carbon sources, the protein gets degraded in stationary phase, during nitrogen starvation, or rapamycin treatment. The latter two are accordance with previous observations (Levchenko et al, Plos one, 2016). Surprisingly, and in contrast to this published work, the authors determine that protein destabilization is mediated by the proteasome and not by vacuolar degradation. Using a purification/mass spec approach, they identify a potential ubiquitination site on Atg32. Finally, the authors mutagenize the site, finding a subtle stabilization of the protein and, correlatively, a subtle (but statistically insignificant) enhancement in mitophagy. This study is of potential interest because it suggests a balance between Atg32 function and its degradation by the proteasome may fine-regulate mitophagy, which indeed warrants further exploration. However, the results are in direct conflict with published observations, which is not discussed and should be addressed. Additionally, the effects with proteasomal inhibition are only convincingly observed after extended periods of time, raising the concern that the stabilization of Atg32 seen is indirectly due to a stress response rather than acute inactivation of Atg32 degradation. Further, the data suggesting that Atg32 is ubiquitinated in a regulated manner at the K282 site are based on an unconvincing purification approach.

Specific points:

1. Stabilization of Atg32 upon proteasome inhibition is only observed after 24h of treatment in stationary phase growth. The data shown in Fig. 4B after shorter periods of treatment are not quantified or convincing. This raises the concern that the authors are not observing a prevention of the typical Atg32 degradation, but rather a potential cell stress response or adaptation. Under nitrogen starvation conditions, the stabilization the authors claim exist at 3h is unconvincing, and there is no effect during longer term treatment.

The authors also examine Atg32 stability during MG132/rapamycin treatment (Fig. S3), and this time find a compelling effect. However, this is several hours after treatment and also in conflict with data from Levchenko et al (who treated with MG132 for 2h and saw no stabilization of Atg32). This needs to be discussed/addressed by the authors.

2. The data identifying a ubiquitinated form of Atg32 are lacking key controls. The authors purify Atg32, and perform a western blot with ubiquitin. However, there is no western blot indicating that the Coomassie stained band is indeed the purified protein. No input is shown from the purification. This is all particularly concerning given that the protein is supposedly SDS resistant after purification, and no protein is observed at the native size. The authors do identify one single peptide of ubiquitinated protein, which they map to K282, however, it is impossible to assess if this is biologically relevant as the mutagenized protein has a negligible effect.

3. It is not clear whether quantification of all western blots is normalized to Pgk1, however this is required. The authors state in the legends, “Atg32-V5 expression was quantified as the percentage of Atg32-V5 level of T0”.

4. The clarity of the manuscript needs to be improved. The authors should explicitly state growth conditions of their assays. It is not obviously stated that “growth” means allowing cultures to reach saturation after dilution to exponential phase. Specific growth media are not always mentioned. Another example of confusion is the Idp1 degradation assay – while this may be standard for the field, the relationship between its cleavage as an indicator of mitophagy is not explicitly stated.

Reviewer #2: In this manuscript, the authors found that the mitophagy receptor Atg32 is degraded upon mitophagy induction or under nitrogen starvation conditions. They further showed that the degradation of Atg32 is mediated by the 26S proteasome but not by autophagy. Using mass spectrometry analysis, they identified Lys 282 is an ubiquitination site on Atg32. Intriguingly, the replacement of Lys 282 with Ala did not slow down the degradation rate of Atg32 in their experimental conditions. Overall, the authors concluded that the yeast mitophgy receptor Atg32 can be ubiquitinated and degraded by the proteasome.

Based on the presented data, this reviewer think this manuscript is not technically sound and many critical controls are missing (see below for details). The quantification data are confused, as some blots contain 5 repeats whereas some blots in the same experiments contain 6 repeats (for example, 8 hrs vs. 24 hrs in Fig. 2B and Fig. 3B). Most quantification assays lack statistical analyses. The writing basically sounds good, but typos and errors avoid the reviewer to fully understand the context.

Major problems:

1. The Atg32 degradation assay were performed by using an epitopic-tagged Atg32 construct instead of integrated tagged Atg32, which makes the full story less convincing. This reviewer suggest the authors at least re-examine the stability and ubiquitination of Atg32 using integrated form.

2. In Figure 1A, the authors should examine the localization of Atg32-V5 protein by immunoflorescent staining instead of gradient assay alone. Co-fraction of two protein does not necessarily mean that they spatially localized together. Moreover, in previous results, Dr. Koji Okamoto showed that the C-terminal tagging in Atg32 would disrupt mitochondrial targeting and out-membrane insertion. Therefore, they generated an internally 3xHA-tagged Atg32 variant (Atg32-3HAn) to replace the endogenous protein. This is the major problem in this manuscript.

3. CHX assay is critical to confirm the decrease of Atg32 over the time (upon to 48 hrs) under mitophagy induction is due to protein degradation but not the synthesis of mRNA transcripts. Unfortunately, the whole Figure 4 are somehow disrupted and this reviewer can not see it.

4. In Figure 3, the ubiquitination levels of Atg32 should be examined by immunoprecipitation of Atg32. The ubiquitination levels of total proteins does not mean Atg32 is also ubiquitinated. In addition, PMSF is unstable in aqueous solutions, and its half-life is about 30 min. How the authors exclude the possibility that less protein levels of Atg32 is due to no effect of PMSF after 24 or 48 hours incubation.

5. In Figure 5A, the strain source of pre2-2 should be referred. The authors should clearly describe how they inactivate 20S proteasome using this mutant strain. The positive control is missing in 5A and 5C.

6. In Figure 6D, ATG32 deletion strain should be included as a negative control.

7. From Figure 7 to 10, these data may remove to the supplemental data since they are negative results and the model figure is not closely related to the main conclusion.

8. The discussion part should be more concise.

Minor issues:

1. Page 8, line 170, the font of “107” is weird.

2. Page 11, line 221, remove “we”

3. Page 14, line 291, the blue label should be removed.

4. Page 31, line 709, where shows “against histidine”?

5. Figure 7B, “imput” should be “input”.

6. PLOS authors have the option to publish the peer review history of their article (what does this mean?). If published, this will include your full peer review and any attached files.

Reviewer #1: No

Reviewer #2: No

---

## [Author Response · Author response to Decision Letter 0]

1 Sep 2020

Reviewer's Responses to Questions

Comments to the Author

1. Is the manuscript technically sound, and does the data support the conclusions?

The manuscript must describe a technically sound piece of scientific research with data that supports

the conclusions. Experiments must have been conducted rigorously, with appropriate controls,

replication, and sample sizes. The conclusions must be drawn appropriately based on the data

presented.

Reviewer #1: No

Reviewer #2: No

To our knowledge, no specific antibodies against the Atg32 protein are available so far. We

agree that labeling the protein at the locus would have been the ideal construct for every study aiming

to examine stability and expression of the Atg32 protein. However, we believe that the construct we

used in our study to express Atg32 protein with a V5-6HIS label at its C-terminus is a good

compromise. We assert that the ATG32 gene in our construct has been put under the control of its own

promoter. This type of strategy has been used in many published works. Moreover, we did not limit

ourselves to testing a single construct; we also used the plasmid provided by Dr. Klionsky (Michigan

University, USA) which allows the expression of the N-terminal labeled Atg32 protein with the HA tag.

We obtained the same results using both plasmids.

Also, we reversed the phenotype of atg32Δ mutant expressing Atg32-V5 plasmid; the protein is

localized into mitochondria (Fig 1A), and we also included figure S1A to demonstrate mitochondrial

localization of Atg32-V5 using purified mitochondria isolated from atg32Δ mutant cells expressing

Atg32-V5 harvested in the mid-exponential phase. Additionally, we demonstrated that Atg32-V5

restored mitophagy in atg32Δ mutant cells to the level similar in wild-type cells in stationary phase

(Fig 1B) as well upon starvation (Fig S1B).

Further, to confirm the significance of our findings, we used a proteasome mutant pre2-2 viable

in our experimental setup as well as a vacuolar protease mutant pep4D. We believe our results provided

clear evidence that the described changes in stability/turnover of Atg32 protein are not due to

unspecified/long-term stress as suggested by the reviewer. In our study, we focused mainly on the two

most physiological conditions for inducing mitophagy: stationary growth phase and nitrogen

deficiency, rather than the use of rapamycin, which is easier to use but causes more stress to the cells.

We believe that we used proper control in each experiment. All experiments were performed

independently several times, as indicated for each experiment. The sample sizes in the same type of

experiments are identical as described in the Material and Methods section/Figure legends in detail.

2. Has the statistical analysis been performed appropriately and rigorously?

Reviewer #1: Yes

Reviewer #2: No

All experiments were, as indicated for each experiment in Material and Methods section,

performed several times, between 3 and 8 depending on the individual experiment. Not all experiments

were repeated the same number of times; however, each tested/examined condition was repeated

independently at least 3 times.

To present our results, we chose a form that allows us to show the values obtained in each

experiment instead of the more common way of displaying a bar graph with standard deviation. In our

chosen form, it is then possible in some cases to see a different number of points that reflect the

different number of independent experiments performed for that experiment.

P-values were calculated using unpaired Students’ t-tests; * P < 0.05 or ** P < 0.01 were

considered statistically significant. In general, we believe that we performed the statistical analysis

appropriately. However, we apologize for not being completely rigorous and not including statistical

analysis for each result in the previous manuscript. We tried to address this problem in our current

revised version.

3. Have the authors made all data underlying the findings in their manuscript fully available?

The PLOS Data policy requires authors to make all data underlying the findings described in their

manuscript fully available without restriction, with rare exception (please refer to the Data Availability

Statement in the manuscript PDF file). The data should be provided as part of the manuscript or its

supporting information, or deposited to a public repository. For example, in addition to summary

statistics, the data points behind means, medians and variance measures should be available. If there

are restrictions on publicly sharing data—e.g. participant privacy or use of data from a third party—

those must be specified.

Reviewer #1: Yes

Reviewer #2: No

With our original submission, we also provided a PDF file entitled “Compilation of original

blots” with all original blots presented in the manuscript. It also is available with this revised version.

With the current version, we also provide a document with the data used for the graphs.

4. Is the manuscript presented in an intelligible fashion and written in standard English?

PLOS ONE does not copy edit accepted manuscripts, so the language in submitted articles must be

clear, correct, and unambiguous. Any typographical or grammatical errors should be corrected at

revision, so please note any specific errors here.

Reviewer #1: Yes

Reviewer #2: Yes

5. Review Comments to the Author

Reviewer #1: In this manuscript, Camougrand et al examine the regulation of the yeast selective

autophagy receptor, Atg32. The authors observe that while Atg32 is specifically expressed during

growth in non-fermentable carbon sources, the protein gets degraded in stationary phase, during

nitrogen starvation, or rapamycin treatment. The latter two are according to previous observations

(Levchenko et al, Plos one, 2016). Surprisingly, and in contrast to this published work, the authors

determine that protein destabilization is mediated by the proteasome and not by vacuolar degradation.

Using a purification/mass spec approach, they identify a potential ubiquitination site on Atg32. Finally,

the authors mutagenize the site, finding a subtle stabilization of the protein and, correlatively, a subtle

(but statistically insignificant) enhancement in mitophagy. This study is of potential interest because it

suggests a balance between Atg32 function and its degradation by the proteasome may fine-regulate

mitophagy, which indeed warrants further exploration. However, the results are in direct conflict with

published observations, which is not discussed and should be addressed.

The majority of the data we obtained do not conflict with the data in the literature. For example,

the decrease in the level of Atg32 protein in the stationary phase was observed, among others, by the

teams of Dr. Ohsumi and Dr. Levchenko. We are familiar with Dr. Levchenko et al.’s (2016) results

published in PlosOne (2016).

We have been working on the study of mitophagy for many years and were the first to show the

selectivity of the process in 2004 (Kissova et al., JBC) and 2007 (Kissova et al., Autophagy). More

importantly, we described the condition in which mitophagy can be observed, and this finding has since

served as an essential tool for all research in this area. Namely, to be able to observe mitophagy, yeast

cells have to be grown in strict respiratory conditions that allow cells to develop fully differentiated

mitochondria and the metabolisms that depend on mitochondria. There is a huge difference in

mitochondrial metabolism between the cells grown in media supplemented with a respiratory carbon

source (e.g., lactate) or with a fermentative carbon source (e.g., glucose).

In his work, Levchenko et al. (2016) used lactate, a respiratory carbon source, which is identical

we use. But there is one essential difference between their and our experimental setup: while we shift

cells first cultivated in a medium with lactate to nitrogen starvation medium with lactate, for the

mitophagy induction, Levchenko et al. used a nitrogen starvation medium supplemented with glucose.

In our setup, cells starved for nitrogen in the presence of a respiratory carbon source, while in

Levchenko et al.’s setup, the cells’ metabolisms are shifted to a different condition. With a shift from

lactate to glucose, not only are mitochondria not needed to the extent they are in respiratory condition,

but the quantity of mitochondria is also regulated by the process of mitochondrial biogenesis. In

accordance with some published results (e.g., Kanki at al, 2009), our unpublished results also showed

that in normal growth conditions (in absence of starvation) a simple shift of cells from respiratory

condition (lactate) to fermentative condition (glucose) does not induce mitophagy, although the

quantity of mitochondria is drastically reduced following such shift in a very short time.

Based on our 40 years of experience with yeast mitochondrial metabolism, it is very difficult to

compare the results obtained in two such diverse experimental setups (Levchenko et al.’s versus ours).

It is easily imaginable that cells have or developed various mechanisms for regulation stability and

quantity of Atg32 in these different conditions. One can be more dependent on vacuolar degradation

while another can undergo proteasomal degradation control.

Moreover, we did not find data on whether Levchenko’s nitrogen starvation medium was

buffered to pH5.5, which we do, and it is important.

Additionally, the effects with proteasomal inhibition are only convincingly observed after extended

periods of time, raising the concern that the stabilization of Atg32 seen is indirectly due to a stress

response rather than acute inactivation of Atg32 degradation.

In our opinion, our results demonstrated a convincing and unambiguous effect of proteasomal

inhibition on the stabilization of Atg32 protein in normal growth conditions.

Let us explain our statement/results in more detail.

In our study, we used the proteasomal inhibitor, MG-132, in two different types of experiments:

First, we examined levels of Atg32 protein in course of the cell growth. We noticed the apparent

decrease in Atg32 levels when cells are entering into a stationary phase of growth—at the time when

mitophagy is induced (Fig. 2). In our experiments, we use control cells (T0) harvested in the midexponential

growth phase that represents the optical density around 1.5–2 OD600. In order to further

elucidate the role of Atg32 in mitophagy and regulation of ATG32 expression, we treated the cells with

the proteasomal inhibitor. MG-132 was added into cell culture at T 8 h (the late-exponential phase).

Cells grew further in inhibitor presence for additional 16 h (early stationary phase, 24 h) or 40 h (late

stationary phase; 48 h). Further, we checked the inhibitor’s effect on the viability of cells (Fig. S5A) to

ensure cells are not affected in our experimental conditions, because longer incubation time could cause

additional stress for cells as the reviewer pointed out. Also, we used time periods of 24 h and 48 h

because those are commonly used to examine mitophagy during normal cell growth/stationary phase.

Although mitophagy can be induced slightly before cells enter into the stationary phase of growth, its

levels are very low before the 24-hour point and cannot be reproducibly detected by immunoblotting

using a standard mitophagy test following the Idp–GFP processing tool that is based on releasing a free

GFP form. In our work, during normal growth conditions, we did not present results from shorter than

24 hours in the presence of inhibitors, so we are not sure what led the reviewer to raise a concern that

stabilization of Atg32 occurs only after long incubation times and may be an indirect effect. The graph

in Figure 3B includes T8 for control conditions, but it does not show point T8 plus MG-132 because

the inhibitor was only added into the medium at that moment. At the 24 h (as well 48 h) time point, the

effect of proteasome inhibition on Atg32 stability is unequivocal, as you can see in Figure 3AB.

Second, after we observed the effect of proteasome inhibition on the Atg32 levels in cells in the

stationary phase, we assessed the alteration of Atg32 protein levels in wild-type yeast cells that were

grown in control media and later treated with cycloheximide to turn off protein expression.

Cycloheximide was added in both the presence and the absence of MG-132, and the positive effects on

stability of Atg32 protein were already observed after a short time of MG-132 treatment—after 20, 40,

or 60 min - Fig. 4B,C.

We think that these results reliably dispel the reviewer’s concern that the stabilization of Atg32

seen is indirectly due to a stress response rather than acute inactivation of Atg32 degradation.

On the other hand, it is important to note that we used two conditions to induce mitophagy:

stationary phase and nitrogen deficiency. We no longer want to use rapamycin because side effects on

various cell functions are significant, and certainly rapamycin treatment can represent stress for cells.

Further, the data suggesting that Atg32 is ubiquitinated in a regulated manner at the K282 site are

based on an unconvincing purification approach.

Could the reviewer kindly specify his reservation regarding his opinion “an unconvincing

purification approach? We believe we follow a standard protocol and purification as described in detail

in the Material and Methods section. At the end of purification, all fractions absorbing at 254 nm were

pooled and analyzed by electrophoresis on 11% SDS-PAGE and submitted to blue colloidal staining

(Fig. 7A: marked as Eluted fraction) and immunoanalysis using antibodies against HIS tag (to detect

Atg32 protein) and antibodies against ubiquitin (Fig. 7B). We would like to note we modified Figure

7B to make this clearer, more details were also added into figure legend. Into our current revised

version, we also included supplemental Fig. S7 that describes purification and detection of Atg32-V5

more in detail.

We agree that the shift in mobility of Atg32 protein after purification is surprising but not

impossible to understand. Atg32 is a protein inserted into the outer membrane of the mitochondria. It

has already been observed and published that some mitochondrial membrane proteins, such as e.g. the

ATP synthase subunit 9 is mainly found in the form of oligomers in SDS-PAGE gel, despite the

presence of detergents in the gel. (FYI: (ref 1) Organization of the yeast ATP synthase F0: a study

based on cysteine mutants, thiol modification and cross-linking reagents Jean Velours, Patrick Paumard

Vincent Soubannier Christelle Spannagel Jacques Vaillier Geneviève Arselin Pierre-Vincent Graves

Biochimica and Biophysica Acta 1458 (2000) 443-456 or (ref 2) ATP Synthase of Yeast Mitochondria

Isolation of the subunit h and disruption of the ATP14 gene * Geneviève Arselin, Jacques Vaillier,

Pierre-Vincent Graves and Jean Velours ‡ JBC vol. 271, No. 34, Issue of August 23, pp. 20284– 20290

1996). The second reference would provide better insight into the problem.

Regarding Levchenko et al article, we understand that it underwent the reviewing process and is

now accepted as a “base” for new findings, but the results presented in their article raises several

questions for us, such as these:

(i) In Figure 2, a band appears around 100 kDa in the delta pep4 strain after treatment with rapamycin.

In Figure 3B, in the delta pep4 delta atgx double mutants treated with rapamycin (time is not

indicated), the band is at 100 kDa plus another band which appears above whose size is not indicated.

(ii) In Figure 3C, a band is indicated by an arrow, but the size is not mentioned.

(iii) In Figure 4A and 4C, after treatment with rapamycin, there is a band at 100 kDa + 1 band between

75 and 100 kDa. It should also be noted that the band revealing Atg32 is wide and diffuse. Could these

additional bands not correspond to protein aggregates containing Atg32 as revealed by western blots

and which would form more in the delta pep4 strain because they are deficient in vacuolar proteases?

Why would it not be due to the stress of “rapamycin + absence of vacuolar proteases?”

To conclude, the data from Levchenko et al. are disturbing: depending on the gels, different

bands are revealed: a band between 75 and 100 kDa, one band at 100 kDa and, in some cases, a band of

high molecular weight greater than 100 kDa

In addition, it has recently been shown that other Atg proteins are also ubiquitinated; this is the case for

Atg9 and Atg14 (Hu et al, 2020, BBRC). The Atg32 protein would be added to this list.

Specific points:

1. Stabilization of Atg32 upon proteasome inhibition is only observed after 24h of treatment in

stationary phase growth. The data shown in Fig. 4B after shorter periods of treatment are not

quantified or convincing. This raises the concern that the authors are not observing a prevention of the

typical Atg32 degradation, but rather a potential cell stress response or adaptation. Under nitrogen

starvation conditions, the stabilization the authors claim exists at 3h is unconvincing, and there is no

effect during longer term treatment.

The experiments presented in Figure 3A and Figure 4B do not give the same information.

Figure 4B gives us information on the half-life of the Atg32 protein in the exponential phase. The

addition of cycloheximide makes it possible to block the synthesis and to see how long the protein is

stable. The results allow us to say that its half-life is shorter than that of a cytosolic (Pgk1) and

mitochondrial (porin) protein. Degradation by the proteasome is involved in Atg32 turnover.

In Figure 3A, MG-132 is added after the 8 h time point (late exponential phase) when cell growth

slows down to enter the stationary phase. The addition of MG-132 at the early- or mid-exponential (T0)

phase of growth is not possible because there would be too much stress for the cells to sustain

inhibition of the proteasome along with the growth.

We re-arranged the order of figures in revised manuscript to make this point clearer.

The authors also examine Atg32 stability during MG132/rapamycin treatment (Fig. S3), and this time

find a compelling effect. However, this is several hours after treatment and also in conflict with data

from Levchenko et al (who treated with MG132 for 2h and saw no stabilization of Atg32). This needs to

be discussed/addressed by the authors.

The only explanation we can offer is that a difference in strains and growing conditions leads to

the involvement of different pathways regulating stability/expression of Atg32. Based on our

experiences, that would not be so surprising. In addition, we used longer rapamycin treatment times.

However, in the study by Levchenko et al., the authors do not mention that they observe the

same level of Atg32 protein in the wild-type and pep4D strains (see Fig. 3 and Fig. 4 in their study).

We have obtained the same result in our work. Moreover, in the figure legends of their paper, they do

not mention in which medium the cells grew.

2. The data identifying a ubiquitinated form of Atg32 are lacking key controls. The authors purify

Atg32, and perform a western blot with ubiquitin. However, there is no western blot indicating that the

Coomassie stained band is indeed the purified protein. No input is shown from the purification. This is

all particularly concerning given that the protein is supposedly SDS resistant after purification, and no

protein is observed at the native size.

We have modified Figure 7 to show that the bands revealed by the colloidal blue staining were

also detected by anti-HIS and anti-Ub antibodies. Both bands, B1 and B2, were cut and analyzed by

mass spectrometry. The Atg32 protein was found in band B1; band B2 corresponds to a histidine-rich

protein, which was also found ubiquitinated.

For more details, please, see also our answer for a similar question of the first reviewer above.

The authors do identify one single peptide of ubiquitinated protein, which they map to K282, however,

it is impossible to assess if this is biologically relevant as the mutagenized protein has a negligible

effect.

The Atg32 protein has 43 lysines, but the analysis covered only 17 lysines; 26 lysines remain

for study. We added more details in the manuscript.

3. It is not clear whether quantification of all western blots is normalized to Pgk1, however this is

required. The authors state in the legends, “Atg32-V5 expression was quantified as the percentage of

Atg32-V5 level of T0”.

Thank you for the comment. We modified all the graphs following the amount of Atg32-V5

protein (namely figures 2B, 2D, 3B, 3D, 4C, 5B, 8C, S2B, S3B and S4C) - quantification of all western

blots is now normalized to Pgk1.

4. The clarity of the manuscript needs to be improved. The authors should explicitly state growth

conditions of their assays. It is not obviously stated that “growth” means allowing cultures to reach

saturation after dilution to exponential phase.

Thank you for the comment. We have revised the Material and Methods section, added more

information to improve the clarity.

Specific growth media are not always mentioned.

The growth conditions are indicated in each legend of the figures and are always the same -

complete minimal synthetic medium with lactate as carbon source (CMS-L; described in the Material

and Methods section in detail), except for Figures 5A and B, where galactose was used as a carbon

source (CMS-G) because the mutant pre2-2 does not grow in media supplemented with respiratory

carbon source - lactate. This is also explained in the Figure legends as well as in the Results part.

Another example of confusion is the Idp1 degradation assay – while this may be standard for the field,

the relationship between its cleavage as an indicator of mitophagy is not explicitly stated.

Thank you for the comment. It is a test commonly used in all studies focusing on mitophagy in

yeast. We modified the text and described the method following the processing of the Idp-GFP protein

as an indicator of mitophagy induction, the reference and description are also provided in the Material

and Methods section.

Reviewer #2: In this manuscript, the authors found that the mitophagy receptor Atg32 is degraded

upon mitophagy induction or under nitrogen starvation conditions. They further showed that the

degradation of Atg32 is mediated by the 26S proteasome but not by autophagy. Using mass

spectrometry analysis, they identified Lys 282 is an ubiquitination site on Atg32. Intriguingly, the

replacement of Lys 282 with Ala did not slow down the degradation rate of Atg32 in their experimental

conditions. Overall, the authors concluded that the yeast mitophgy receptor Atg32 can be ubiquitinated

and degraded by the proteasome.

Based on the presented data, this reviewer think this manuscript is not technically sound and many

critical controls are missing (see below for details). The quantification data are confused, as some

blots contain 5 repeats whereas some blots in the same experiments contain 6 repeats (for example, 8

hrs vs. 24 hrs in Fig. 2B and Fig. 3B). Most quantification assays lack statistical analyses. The writing

basically sounds good, but typos and errors avoid the reviewer to fully understand the context.

Indeed, we accumulated more data at 24 h and 48 h of growth because these were the most

important conditions for our study. The 8 h point was not always taken, which explains the difference

in points. We have chosen this presentation for the graphs to show all the data obtained. This question

could have been avoided if we had presented the results in the form of bars.

Major problems:

1. The Atg32 degradation assay were performed by using an epitopic-tagged Atg32 construct instead of

integrated tagged Atg32, which makes the full story less convincing. This reviewer suggest the authors

at least re-examine the stability and ubiquitination of Atg32 using integrated form.

As we already mentioned above, to our knowledge, no specific antibodies against the Atg32

protein are available so far. We agree that labeling the protein at the locus would have been the ideal

construct for every study aiming to examine stability and expression of the Atg32 protein. However, we

believe that the construct we used in our study to express Atg32 protein with a V5-6HIS label at its Cterminus

is a good compromise. We assert that the ATG32 gene in our construct has been put under the

control of its own promoter. This type of strategy has been used in many published works.

2. In Figure 1A, the authors should examine the localization of Atg32-V5 protein by immunoflorescent

staining instead of gradient assay alone. Co-fraction of two protein does not necessarily mean that they

spatially localized together. Moreover, in previous results, Dr. Koji Okamoto showed that the Cterminal

tagging in Atg32 would disrupt mitochondrial targeting and out-membrane insertion.

Therefore, they generated an internally 3xHA-tagged Atg32 variant (Atg32-3HAn) to replace the

endogenous protein. This is the major problem in this manuscript.

We have chosen a biochemical approach to look at the localization of the Atg32 protein, which

is as relevant as fluorescence experiments. Using gradients to fractionate cell extracts has long been

common. The results show that the Atg32 protein is found in the same fraction as the porin that is

localized to the mitochondrial outer membrane when study was performed on total cell lysate (Fig.

1A). In new supplementary figure 1A, we also isolated mitochondria that we loaded on a sucrose

gradient - Atg32-V5 protein was found in the same fractions as porin.

3. CHX assay is critical to confirm the decrease of Atg32 over the time (upon to 48 hrs) under

mitophagy induction is due to protein degradation but not the synthesis of mRNA transcripts.

Unfortunately, the whole Figure 4 are somehow disrupted and this reviewer can not see it.

Perhaps we did not fully clarify our rationale for the experiment with cycloheximide which was

to check the half-life of the Atg32 protein. We modified the text in the manuscript to explain the

purpose of this experiment better.

4. In Figure 3, the ubiquitination levels of Atg32 should be examined by immunoprecipitation of Atg32.

The ubiquitination levels of total proteins does not mean Atg32 is also ubiquitinated.

We agree with this comment. However, the purpose of the ubiquitin blot was to control and

show the effect of MG-132 as an effective inhibitor of proteasome in our experimental setup. We added

an explanation in the manuscript.

In addition, PMSF is unstable in aqueous solutions, and its half-life is about 30 min. How the authors

exclude the possibility that less protein levels of Atg32 is due to no effect of PMSF after 24 or 48 hours

incubation.

We agree with the comment. We added PMSF several times during the experiment, and we

added this information in the manuscript.

In addition, we used the pep4Δ mutant lacking vacuolar proteases and obtained the same results as we

obtained with the use of a vacuolar protease inhibitor (PMSF).

5. In Figure 5A, the strain source of pre2-2 should be referred. The authors should clearly describe how

they inactivate 20S proteasome using this mutant strain.

Dr. Sagot kindly contributed the pre2-2 mutant. We added this information in the text.

Pre2 protein is the β5 subunit of the 20S proteasome and is responsible for the chymotryptic activity of

the proteasome. We added this information in the text.

The positive control is missing in 5A and 5C.

For us, the positive controls were the two strains, atg32Δ + Atg32-V5 and BY4742+ Atg32-V5.

6. In Figure 6D, ATG32 deletion strain should be included as a negative control.

Like processing of the Idp-GFP tool for detection of mitophay by western blot technique, the

mt-PHO8 ALP reporter test is usually used as a biochemical approach to study mitophagy in yeast. To

our knowledge, including the atg32Δ mutant strain (defective for mitophagy) as a negative control is

not always required.

However, we included for the reviewer a figure to show that mitophagy (mtALP activity) is not

induced/elevated over basal level (T0 wild type - 1st column; T0 mutant- 3rd column) during starvation

in atg32Δ mutant (-N6h mutant - 4th column) as compared to wild type where mtALP increased about

3 times at the same time (-N6h wild type - 2nd column). We performed this experiment under the same

conditions as described in Fig. 6D.

7. From Figure 7 to 10, these data may remove to the supplemental data since they are negative results

and the model figure is not closely related to the main conclusion.

While we understand your concern, even if they are negative results, we would like to keep

them among the main results, in accordance with PLOS policy.

8. The discussion part should be more concise.

We modified the Discussion part of the manuscript at places where it seemed to help the clarity

of the manuscript. To improve the clarity of the manuscript, we modified Results, Material and

Methods, figures substantially.

Minor issues: were corrected as needed

1. Page 8, line 170, the font of “107” is weird.

2. Page 11, line 221, remove “we”

3. Page 14, line 291, the blue label should be removed.

4. Page 31, line 709, where shows “against histidine”?

5. Figure 7B, “imput” should be “input

One additional figure for reviewers has been added in the cover letter

---

## [Decision Letter · Decision Letter 1]

17 Sep 2020

PONE-D-20-11526R1

The yeast mitophagy receptor Atg32 is ubiquitinated and degraded by the proteasome

PLOS ONE

Dear Dr. Camougrand,

Thank you for submitting your manuscript to PLOS ONE. After careful consideration, we feel that it has merit but does not fully meet PLOS ONE’s publication criteria as it currently stands. Therefore, we invite you to submit a revised version of the manuscript that addresses the points raised during the review process.

One expert reviewer has returned comments on your resubmission. This reviewer still has concerns about the manuscript that should be addressed by further revision. Specifically, you must address concerns regarding the different K282A clones and their differential behavior. Furthermore, you should also address the reviewer's query about MG-132 treatment and, if additional experiments are not forthcoming, you should provide all necessary caveats within the text of your manuscript.

We look forward to receiving your revised manuscript.

Kind regards,

Cory D. Dunn, Ph.D.

Academic Editor

PLOS ONE

Reviewers' comments:

Reviewer's Responses to Questions

**Comments to the Author**

1. If the authors have adequately addressed your comments raised in a previous round of review and you feel that this manuscript is now acceptable for publication, you may indicate that here to bypass the “Comments to the Author” section, enter your conflict of interest statement in the “Confidential to Editor” section, and submit your "Accept" recommendation.

Reviewer #1: (No Response)

2. Is the manuscript technically sound, and do the data support the conclusions?

Reviewer #1: Yes

3. Has the statistical analysis been performed appropriately and rigorously? 

Reviewer #1: I Don't Know

4. Have the authors made all data underlying the findings in their manuscript fully available?

Reviewer #1: Yes

5. Is the manuscript presented in an intelligible fashion and written in standard English?

Reviewer #1: No

6. Review Comments to the Author

Reviewer #1: In the revised manuscript by Camougrand et al, the authors made modest changes to address concerns raised in their original submission. While I have my doubts that their purification strategy is effectively working (ie, a concern is that the oligomerized band is non-specifically cross-reacting with the anti-his antibody), the authors have now provided the requested data. However, it is somewhat glaring that the authors have now removed data from Figure 8 related to the ubiquitinated band they identified via proteomic analysis. They had previously generated two clonal lines of a K282A mutant, one of which showed no change in Atg32 turnover and the other which mildly stabilized Atg32 at the 8h timepoint. In the revised manuscript, the “negative” data is removed while “significantly (P<0.05) impaired” is added to the text. The authors should return “clone 1” to the manuscript for transparency purposes and graphically display the values that led to this conclusion.

My other primary concern regarding lengthy treatment with MG-132 was not addressed. In their response, the authors write that “we did not present results from shorter than 24 hours in the presence of inhibitors, so we are not sure what led the reviewer to raise a concern that stabilization of Atg32 occurs only after long incubation times and may be an indirect effect.” This is exactly the point – because the authors do not examine other time points (for example, 1h, 2h, 4h) after MG-132 addition at 8h, they can only state that 16h of treatment leads to stabilization of Atg32 during stationary growth. While the authors now demonstrate that such treatment does not inhibit cell growth, they also cannot state whether the protein is acutely stabilized or stabilized as part of an adaptive cellular response to prolonged proteasomal inhibition. In exponentially growing cells, where there is no appreciable mitophagy, the authors use short-term MG132 treatment in combination with cycloheximide and observe a very modest inhibition of proteasomal turnover. So while the authors can conclude that Atg32 can be targeted to the proteasome during exponential growth, it is not clear that this is an acute cellular response to regulate mitophagy when cells reach stationary phase. If the authors are unwilling to perform new experiments, they must substantially improve the clarity of the manuscript and acknowledge the potential caveats and alternative explanations of their results.

7. PLOS authors have the option to publish the peer review history of their article (what does this mean?). If published, this will include your full peer review and any attached files.

Reviewer #1: No

---

## [Author Response · Author response to Decision Letter 1]

2 Oct 2020

Answers to reviewer 1

Reviewer #1: In the revised manuscript by Camougrand et al, the authors made modest changes to

address concerns raised in their original submission. While I have my doubts that their purification

strategy is effectively working (ie, a concern is that the oligomerized band is non-specifically crossreacting

with the anti-his antibody), the authors have now provided the requested data.

However, it is somewhat glaring that the authors have now removed data from Figure 8 related to

the ubiquitinated band they identified via proteomic analysis. They had previously generated two

clonal lines of a K282A mutant, one of which showed no change in Atg32 turnover and the other

which mildly stabilized Atg32 at the 8h timepoint. In the revised manuscript, the “negative” data is

removed while “significantly (P<0.05) impaired” is added to the text. The authors should return

“clone 1” to the manuscript for transparency purposes and graphically display the values that led

to this conclusion.

We would like to thank you for all your effort and comments. We believe they helped us to

make our manuscript better.

We apologize if the noninclusion in our revised manuscript of the immunoblot result with

clone 1 in Figure 4A gave the impression that we are trying to intentionally select the results that fit

into our story. By no means was this our intent. We only tried to simplify the presentation of the

results and eliminate unnecessary duplication. Let us provide you with a more detailed explanation.

In the original version, we presented in Figure 8A an image of blots for two different clones

(1 and 2). In Figure 8C, there is only one column for the mutant K282A (it does not specify if it is

clone 1 or clone 2). We apologize for not explaining this better in the text—the column (48h)

contains results from 6 independent experiments from 3 different clones that we tested (clone 1 and

clone 2 from Figure 8A plus a third clone that was not mentioned in the manuscript). Also, it has to

be noted that in the original version, the y-axis of the graph in Figure 8C shows the amount of

Atg32-V5 normalized to T0 value of Atg32-V5, which, based on the reviewer's request, was

normalized in the revised version to the amount of Pgk1 at each time.

Again, Figure 8 in the revised version of our manuscript does include the results of 6

independent experiments from 3 individual clones. Because quantification normalized to Pgk1 did

not show significant changes between individual clones (1-2-3), we decided to keep only one of the

clones in part A. When compared to wild-type Atg32 protein, there is about 15% (P<0.05) more

mutant protein in the late stationary phase cells (Fig. 8A, C; 48 h).

For the reviewer, we attach here the immunoblot results obtained from 3 individual clones

with individual quantification for each of them for comparison.

Figure for reviewer only

Figure 8 : first soumission

Figure 8 : second revision

My other primary concern regarding lengthy treatment with MG-132 was not addressed. In their

response, the authors write that “we did not present results from shorter than 24 hours in the

presence of inhibitors, so we are not sure what led the reviewer to raise a concern that stabilization

of Atg32 occurs only after long incubation times and may be an indirect effect.” This is exactly the

point – because the authors do not examine other time points (for example, 1h, 2h, 4h) after MG-

132 addition at 8h, they can only state that 16h of treatment leads to stabilization of Atg32 during

stationary growth. While the authors now demonstrate that such treatment does not inhibit cell

growth, they also cannot state whether the protein is acutely stabilized or stabilized as part of an

adaptive cellular response to prolonged proteasomal inhibition.

In our previous answer we tried to explain to the reviewer that we did not examine the

shorter times (fewer than 16 hours; T24) of treatment with MG-132 because our aim was to study

levels of Atg32 protein at the beginning of and during the stationary phase—the time of cell growth

when mitophagy is induced. We believe that our results provide clear evidence that inhibition of

proteasome activity with MG-132 leads to stabilization of Atg32 during the stationary phase and

that this correlates with an increase in mitophagy activity.

However, to address the reviewer’s comment, we performed an experiment in which MG-

132 was added to the culture at T0 (instead of at T8 as we used during our study) and examined the

levels of Atg32 after 8 hours (exponential phase, no appreciable mitophagy), 24 hours (early

stationary phase, beginning of mitophagy induction), and 48 hours (late stationary phase). The

obtained results were included in the manuscript as supplementary Fig. 4D (for your convenience

the results are attached to this letter as well). As you can see, the level of Atg32p at T8 is about half

compared to the level at T0, and it further decreases to where there is almost no detectable Atg32

when measured at T48. On the contrary, the presence of MG-132 drastically recovers Atg32 levels

in both short (8h) and long (24h and 48h) incubation times. We provide the quantification of the

immunoblot results here as well. We strongly believe these results support the view that the Atg32

protein is acutely stabilized instead of being stabilized as a part of an adaptive cellular response to

prolonged proteasomal inhibition.

New supplementary Figure S4 (with new results in part D - included into manuscript):

Quantification figure S4D- an average from 2 independent experiment (for reviewer only):

In exponentially growing cells, where there is no appreciable mitophagy, the authors use short-term

MG132 treatment in combination with cycloheximide and observe a very modest inhibition of

proteasomal turnover. So while the authors can conclude that Atg32 can be targeted to the

proteasome during exponential growth, it is not clear that this is an acute cellular response to

regulate mitophagy when cells reach stationary phase. If the authors are unwilling to perform new

experiments, they must substantially improve the clarity of the manuscript and acknowledge the

potential caveats and alternative explanations of their results.

Beside our aforementioned explanation, our experiments with cycloheximide (Figs. 4B and

4C) showed that the turnover of the Atg32 protein is extremely rapid compared to other proteins

(porin, Pgk1) and that this turnover involves the activity of the proteasome. This also explains why

the activity of the promoter increases during growth (Fig. 4A). After one hour with the

cycloheximide, the effect of MG-132 is striking and significant (P=0.0146; we included data for

“1h+cycloheximide+MG-132” in Fig. 4C that were missing in the previous version). Respectfully,

we cannot agree with the reviewer’s statement that this change is “very modest.”

Our results support the conclusion that Atg32 can be targeted to the proteasome during

exponential growth as well as favoring an acute cellular response to regulate levels of Atg32 protein

(as an essential mitophagy regulator) when cells reach the stationary phase.

We slightly modified some parts of the Results and Discussion sections to make the text

clearer.

The manuscript was proofread by a native English-speaking person

---

## [Editor Report · Decision Letter 2]

12 Oct 2020

PONE-D-20-11526R2

The yeast mitophagy receptor Atg32 is ubiquitinated and degraded by the proteasome

PLOS ONE

Dear Dr. Camougrand:

Thank you for your manuscript resubmission. I believe that the reviewer comments were addressed appropriately in your latest version, and I do not currently plan on sending the paper again to the reviewers. However, I have received the following guidance from the PLOS ONE staff:

"PLOS ONE now requires that submissions reporting blots or gels include original, uncropped and unadjusted blot/gel image data in addition to complying with our image preparation guidelines described at https://journals.plos.org/plosone/s/figures#loc-blot-and-gel-reporting-requirements. The revised submission should include the raw blot/gel image data for your review, either in Supporting Information or via a public data repository; the Data Availability Statement should indicate where these data can be found. The original blot/gel image data should (1) represent unadjusted, uncropped images, (2) be provided for all blot/gel data reported in the main figures and Supporting Information, and (3) match the images in the manuscript figure(s). If you have any questions or concerns about the blot/gel data, or about the author’s compliance with the journal’s blot/gel reporting requirements, you can raise these in your next decision letter or email us at plosone@plos.org".

Could you please follow up with a new version of the manuscript which complies with these guidelines?

Also, I have the following minor comments on the manuscript:

Page 24: MG-123 instead of MG-132.

Page 25: 'Youle's lab' 'Levine's lab' - please seek a more formal way to address the work of these laboratories.

A marked-up copy of your manuscript that highlights changes made to the original version. You should upload this as a separate file labeled 'Revised Manuscript with Track Changes'.An unmarked version of your revised paper without tracked changes. You should upload this as a separate file labeled 'Manuscript'.

We look forward to receiving your revised manuscript.

Kind regards,

Cory D. Dunn, Ph.D.

Academic Editor

PLOS ONE

---

## [Author Response · Author response to Decision Letter 2]

12 Oct 2020

Answers to reviewer 1

Reviewer #1: In the revised manuscript by Camougrand et al, the authors made modest changes to address concerns raised in their original submission. While I have my doubts that their purification strategy is effectively working (ie, a concern is that the oligomerized band is non-specifically cross-reacting with the anti-his antibody), the authors have now provided the requested data. 

However, it is somewhat glaring that the authors have now removed data from Figure 8 related to the ubiquitinated band they identified via proteomic analysis. They had previously generated two clonal lines of a K282A mutant, one of which showed no change in Atg32 turnover and the other which mildly stabilized Atg32 at the 8h timepoint. In the revised manuscript, the “negative” data is removed while “significantly (P<0.05) impaired” is added to the text. The authors should return “clone 1” to the manuscript for transparency purposes and graphically display the values that led to this conclusion.

We would like to thank you for all your effort and comments. We believe they helped us to make our manuscript better. 

 We apologize if the noninclusion in our revised manuscript of the immunoblot result with clone 1 in Figure 4A gave the impression that we are trying to intentionally select the results that fit into our story. By no means was this our intent. We only tried to simplify the presentation of the results and eliminate unnecessary duplication. Let us provide you with a more detailed explanation.

In the original version, we presented in Figure 8A an image of blots for two different clones (1 and 2). In Figure 8C, there is only one column for the mutant K282A (it does not specify if it is clone 1 or clone 2). We apologize for not explaining this better in the text—the column (48h) contains results from 6 independent experiments from 3 different clones that we tested (clone 1 and clone 2 from Figure 8A plus a third clone that was not mentioned in the manuscript). Also, it has to be noted that in the original version, the y-axis of the graph in Figure 8C shows the amount of Atg32-V5 normalized to T0 value of Atg32-V5, which, based on the reviewer's request, was normalized in the revised version to the amount of Pgk1 at each time.

Again, Figure 8 in the revised version of our manuscript does include the results of 6 independent experiments from 3 individual clones. Because quantification normalized to Pgk1 did not show significant changes between individual clones (1-2-3), we decided to keep only one of the clones in part A. When compared to wild-type Atg32 protein, there is about 15% (P<0.05) more mutant protein in the late stationary phase cells (Fig. 8A, C; 48 h). 

For the reviewer, we attach here the immunoblot results obtained from 3 individual clones with individual quantification for each of them for comparison. 

Figure for reviewer only

Figure 8 : first soumission

Figure 8 : second revision

My other primary concern regarding lengthy treatment with MG-132 was not addressed. In their response, the authors write that “we did not present results from shorter than 24 hours in the presence of inhibitors, so we are not sure what led the reviewer to raise a concern that stabilization of Atg32 occurs only after long incubation times and may be an indirect effect.” This is exactly the point – because the authors do not examine other time points (for example, 1h, 2h, 4h) after MG-132 addition at 8h, they can only state that 16h of treatment leads to stabilization of Atg32 during stationary growth. While the authors now demonstrate that such treatment does not inhibit cell growth, they also cannot state whether the protein is acutely stabilized or stabilized as part of an adaptive cellular response to prolonged proteasomal inhibition. 

In our previous answer we tried to explain to the reviewer that we did not examine the shorter times (fewer than 16 hours; T24) of treatment with MG-132 because our aim was to study levels of Atg32 protein at the beginning of and during the stationary phase—the time of cell growth when mitophagy is induced. We believe that our results provide clear evidence that inhibition of proteasome activity with MG-132 leads to stabilization of Atg32 during the stationary phase and that this correlates with an increase in mitophagy activity. 

However, to address the reviewer’s comment, we performed an experiment in which MG-132 was added to the culture at T0 (instead of at T8 as we used during our study) and examined the levels of Atg32 after 8 hours (exponential phase, no appreciable mitophagy), 24 hours (early stationary phase, beginning of mitophagy induction), and 48 hours (late stationary phase). The obtained results were included in the manuscript as supplementary Fig. 4D (for your convenience the results are attached to this letter as well). As you can see, the level of Atg32p at T8 is about half compared to the level at T0, and it further decreases to where there is almost no detectable Atg32 when measured at T48. On the contrary, the presence of MG-132 drastically recovers Atg32 levels in both short (8h) and long (24h and 48h) incubation times. We provide the quantification of the immunoblot results here as well. We strongly believe these results support the view that the Atg32 protein is acutely stabilized instead of being stabilized as a part of an adaptive cellular response to prolonged proteasomal inhibition.

New supplementary Figure S4 (with new results in part D - included into manuscript):

Quantification figure S4D- an average from 2 independent experiment (for reviewer only):

In exponentially growing cells, where there is no appreciable mitophagy, the authors use short-term MG132 treatment in combination with cycloheximide and observe a very modest inhibition of proteasomal turnover. So while the authors can conclude that Atg32 can be targeted to the proteasome during exponential growth, it is not clear that this is an acute cellular response to regulate mitophagy when cells reach stationary phase. If the authors are unwilling to perform new experiments, they must substantially improve the clarity of the manuscript and acknowledge the potential caveats and alternative explanations of their results.

Beside our aforementioned explanation, our experiments with cycloheximide (Figs. 4B and 4C) showed that the turnover of the Atg32 protein is extremely rapid compared to other proteins (porin, Pgk1) and that this turnover involves the activity of the proteasome. This also explains why the activity of the promoter increases during growth (Fig. 4A). After one hour with the cycloheximide, the effect of MG-132 is striking and significant (P=0.0146; we included data for “1h+cycloheximide+MG-132” in Fig. 4C that were missing in the previous version). Respectfully, we cannot agree with the reviewer’s statement that this change is “very modest.” 

Our results support the conclusion that Atg32 can be targeted to the proteasome during exponential growth as well as favoring an acute cellular response to regulate levels of Atg32 protein (as an essential mitophagy regulator) when cells reach the stationary phase. 

We slightly modified some parts of the Results and Discussion sections to make the text clearer. 

The manuscript was proofread by a native English-speaking person.

---

## [Editor Report · Decision Letter 3]

19 Oct 2020

The yeast mitophagy receptor Atg32 is ubiquitinated and degraded by the proteasome

PONE-D-20-11526R3

Dear Dr. Camougrand,

We’re pleased to inform you that your manuscript has been judged scientifically suitable for publication and will be formally accepted for publication once it meets all outstanding technical requirements.

Kind regards,

Cory D. Dunn, Ph.D.

Academic Editor

PLOS ONE
---

## [Editor Report · Acceptance letter]

14 Dec 2020

PONE-D-20-11526R3 

The yeast mitophagy receptor Atg32 is ubiquitinated and degraded by the proteasome 

Dear Dr. Camougrand:

I'm pleased to inform you that your manuscript has been deemed suitable for publication in PLOS ONE. Congratulations! Your manuscript is now with our production department. 

Kind regards, 

on behalf of

Dr. Cory D. Dunn 

Academic Editor

PLOS ONE